# Self-attention-based Diffusion Model for Time-series Imputation in Partial Blackout Scenarios

## Abstract

Missing values are a common phenomenon in multivariate time series data, capable of harming the performance of machine learning models and introducing bias and inaccuracies into further analysis. These gaps typically arise from various sources, including sensor malfunctions, extreme events like blackouts, and human error. Previous work has made promising strides in imputation for time series data. However, they mostly dealt with some selective cases of missing patterns such as - missing at random, missing due to complete blackout (all features are missing for a given period of time), and forecasting. In this paper, we delve into a more general category of missing patterns, which we call **partial blackout**, wherein a subset of features remain missing for one or several consecutive time steps. This describes a more natural scenario that is frequently encountered in real-world applications and covers the aforementioned patterns as special cases. We introduce a two-stage imputation process that explicitly models the feature and temporal correlations with the help of self-attention and diffusion processes. Notably, our model outperforms the state-of-the-art models when dealing with general partial blackout scenarios and exhibits greater scalability, offering promise for practical data imputation needs. The code and the synthetic experiments are here: https://anonymous.4open.science/r/SADI-official-repository-3853/README.md.

## 1 Introduction

Multivariate time-series data frequently exhibit missing values across diverse domains such as finance, meteorology, agriculture, transportation, and healthcare. This common occurrence is attributed to various factors, including but not limited to human fallibility, equipment malfunctions, and suboptimal data input processes (Silva et al., 2012; Yi et al., 2016). Most machine learning algorithms depend on having complete data with no missing values, making imputation an essential tool. It is important to note that suboptimal imputations could have adverse effects like harming the quality of subsequent tasks and potentially introducing bias into further analysis, thereby casting doubt over the integrity of the results (Shadbahr et al., 2022; Zhang et al., 2021).

In the domain of time series data imputation, there have been many deep neural network-based models such as BRITS (Cao et al., 2018), SAITS (Du et al., 2023), GRUI-GAN (Luo et al., 2018), and GP-VAE (Fortuin et al., 2020), etc. In recent years, score-based diffusion models have shown improvement in data generation through a denoising process. They have achieved substantial performance in domains like image generation (Ho et al., 2020; Nichol & Dhariwal, 2021; Song et al., 2020b; Xiang et al., 2023) and audio synthesis (Kong et al., 2020; Chen et al., 2021). Additionally, these models have shown significant gains in performance for imputing missing data and forecasting in the case of time series data. These approaches provide good quality imputations by conditioning on observed values (Tashiro et al., 2021; Alcaraz & Strodthoff, 2022).

However, a recurring theme in many of these works is the attention to limited types of missingness. They focus on scenarios like randomly missing data, missing a feature for a few consecutive time steps (interpolation), and complete blackouts where all features are missing for some time. In our study, we explore a more realistic type of missingness we call "partial blackout," which covers situations where a variable number of features

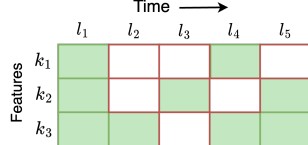

(a) Multiple blocks of 1 feature missing for one or more time steps (Random Missing)

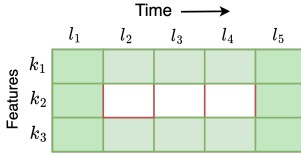

(b) 1 feature missing for more than one time step (Interpolation)

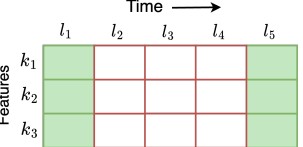

(c) All features missing for one or more time steps (Complete Blackout)

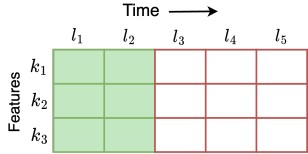

(d) A special case of complete blackout (Forecasting)

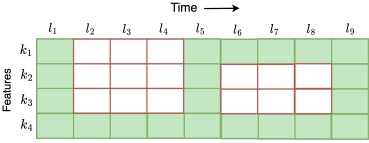

(e) Multiple blocks of 1 or more features missing for one or more time steps (Random Missing)

Figure 1: Partial blackout scenarios

become unavailable for some time. The concept of partial blackouts covers all the other types of missingness, such as randomly missing data, interpolation, complete blackout, and forecasting as special cases.

Importantly, this concept allows multiple distinct blocks of missingness, each characterized by its unique combination of the number of absent features and the duration of the missing time steps, as shown in Figure 1. The random missing scenario in Figure 1a can be expressed as a partial blackout case where multiple single feature blocks are missing for one time step. Similarly, the interpolation (Figure 1b) case can be expressed as a partial blackout with one feature missing for a period of time. It is apparent that the other two scenarios, such as complete blackout (Figure 1c) and forecasting (Figure 1d), are just extreme cases of partial blackouts. This broad and flexible definition of partial blackouts allows us to address a broad spectrum of missingness patterns in real-world datasets.

In the scope of our work, we tackle the challenge of **partial blackout** problem in time series imputation. Our solution, dubbed as **SADI** - short for "**S**elf-**A**ttention-based **D**iffusion Model for Time Series **I**mputation", is a novel approach to diffusion-model-based imputation. The core component of our model is the explicit modeling of feature dependencies across time in the form of "Feature Dependency Encoder" and temporal correlations across features in the form of "Gated Temporal Attention" blocks. We also deploy a two-stage imputation process that considers how the imputed data from the initial stage is further refined and affects the imputation of other data in the second stage. The model combines the results of the two stages as a weighted combination where the weights are learned from the data.

In summary, our contributions are threefold, each adding some depth to the solution of the time-series imputation problem:

1. **Exploration of partial blackout scenario**: We introduce a new archetype of missing patterns called "partial blackout," a more generalized concept encompassing the limited cases of missingness investigated in previous studies and other real-world scenarios.

2. **Explicit modeling of feature and temporal correlations and two-stage imputation**: We introduce a novel diffusion-based imputation model, SADI, which captures feature and time dependencies with explicit components within the denoising function. We also present a two-stage imputation process, where the second stage refines the initial imputations from the first stage, improving the imputation quality.

3. **Empirical study**: We evaluate our model in several real-world time series domains and show that it outperforms and is more scalable than other state-of-the-art models in most partial blackout

scenarios. We employ synthetically designed datasets with various feature dependencies to better understand its strengths and weaknesses.

## 2 Multivariate time-series imputation

We consider multivariate time-series data, which are represented as $X = \{x_{1:L,1:K}\} \in \mathbf{R}^{L \times K}$, where $L$ signifies the time-series length and $K$ denotes the number of features. We assume that the data is distributed according to the joint distribution $\mathbf{P}(X)$. Only a part of the data $X_{obs}$ is observed, and the rest $X_{miss}$ is missing. The problem of multivariate time series imputation is to generate the missing data according to the conditional distribution $\mathbf{P}(X_{miss}|X_{obs})$. We represent the observed and missing data using a binary mask, $M = \{m_{1:L,1:K}\} \in \{0,1\}^{L \times K}$ , where 0 serves as the indicator for the absence of data, while 1 signifies the presence of observations. The training data consists of different samples from the joint distribution $\mathbf{P}$ where some values of $X$ are missing. In our study, we introduce a novel time-series imputation technique that generates the missing values $X_{miss}$ conditioned on the observed values $X_{obs}$ in a test instance.

**Traditional statistical approaches:** Time-series imputation problem has been thoroughly studied and reviewed in both deep learning and non-deep learning community. There have been non-deep learning-based techniques like imputation based on mean/median of the missing features, linear interpolation between nearest observed values and K-nearest neighbor imputation. A more refined approach with good performance is MICE (van Buuren & Groothuis-Oudshoorn, 2011), which uses multiple iterations of linear regression model training to estimate missing values based on the available data from other features.

**Autoregressive models:** Che et al. (2016) propose GRU-D model as a solution for handling missing data in time series classification problems. They introduce the notion of incorporating time decay factor based on the latest observations. This idea has been adopted in subsequent works such as BRITS (Cao et al., 2018) and M-RNN (Yoon et al., 2017), both of which handle missing data with bidirectional RNNs. The BRITS model accounts for correlations among the features, which is absent in M-RNN. BRITS optimizes the reconstruction loss of the observed data along with the consistency of the predicted imputations in the forward and backward directional RNNs.

**Attention-based methods:** There have been some research studies that involve utilizing RNNs with attention mechanisms like GLIMA (Suo et al., 2020) and Shukla & Marlin (2021). Shan & Oliva (2021) introduces a time-series imputation approach, NRTSI, that treats the time-series as a set of (time, data) tuples and employs a transformer encoder to handle irregularly sampled data. Another self-attention-based approach, SAITS, utilizes a two stage self-attention mechanism to capture the temporal dependencies and feature correlations (Du et al., 2023). This method combines the two stage imputation predictions based on the attention maps and missingness information, and jointly optimizes the reconstruction loss of observed data and the imputation loss of artificially omitted data.

**Generative models:** Time-series imputation has also been studied using generative models. These include GAN-based works by Luo et al. (2019; 2018); Liu et al. (2019); Miao et al. (2021) and VAE-based probabilistic imputation methods by Fortuin et al. (2020). These approaches suffer from training instability and fall short of achieving state-of-the-art performance as illustrated by Du et al. (2023). On the other hand, score-based diffusion models, such as CSDI (Tashiro et al., 2021) and SSSD (Alcaraz & Strodthoff, 2022), have demonstrated impressive results and are emerging as strong competitors. The CSDI model separately captures the temporal and feature dependencies with two distinct transformer encoders. It treats each feature similarly when modeling temporal dependencies and each time step similarly when modeling the feature correlations. Another diffusion model, Time-Grad (Rasul et al. (2021)), has demonstrated reliable performance in various forecasting tasks. While Time-Grad excels at predicting future values based on past observations using RNN to handle past time-series data, it cannot leverage the future data for imputation due to its autoregressive nature. There have also been some works based on Bayesian inference modeling. Cui et al. (2019) introduces a novel Bayesian Gaussian Copula Factor (BGCF) approach designed for parameter learning in latent variable models, particularly when dealing with mixed data that includes both continuous and ordinal variables, and also accounts for the presence of missing values. Vidotto et al. (2018) employs a Bayesian approach, incorporating prior beliefs about the parameters into the model; however, it can only handle cross-sectional categorical data, which might limit its applicability to continuous data or mixed data

types without further modifications. Vidotto et al. (2019) proposes an imputation model methodology that involves using multiple imputations under the Missing At Random assumption to replace missing dataset values with plausible ones, generated based on observed data and designed to capture various dependencies and complexities within longitudinal data using Bayesian mixture latent Markov models.

**Other methods:** Other methods include the use of graph neural networks (Cini et al., 2021) to utilize spatial and relational information among input channels; however, the effectiveness of the GRIN model hinges on the assumption that monitored physical quantities can be accurately reconstructed from neighboring sensor observations, requiring a high degree of sensor homogeneity and specific feature availability. A hybrid of latent ordinal differential equation networks and RNN (Rubanova et al., 2019). While the ODE-RNN and Latent ODE models introduced offer significant improvements for modeling irregularly sampled time series data, particularly in handling arbitrary time gaps and enhancing interpretability through continuous-time latent states, they require more computational resources for evaluation and might not scale as effectively with the sparsity of data compared to standard RNN models, as evidenced by the increased evaluation time for ODE-RNNs and Latent ODEs compared to standard GRUs in the experiments. Hyper-parameter optimization to design MLP architecture for long-term time-series imputation (Park et al., 2022) has limitations in the challenge of accurately estimating missing values towards the end of data gaps, where the method occasionally fails, suggesting a potential for error accumulation when predictions are based on previously predicted values. And, provably convergent schrödinger bridge probabilistic imputation (Chen et al., 2023), which is the first work of this kind, but showed worse performance than CSDI (Tashiro et al., 2021).

## 3 Background

In this section we introduce the different mechanisms that form the basis of our proposed approach.

### 3.1 Self-attention mechanism

A self-attention mechanism, introduced by Vaswani et al. (2017), transforms the input sequence into a sequence of vectors that is analyzed and processed further. This process is very effective in detecting patterns and relationships within the input sequence by learning the attention-weights. In self-attention, a given sequence is transformed and mapped into three distinct vectors: a query vector, $q$ (of dimension $d_k$); a key vector, $k$ (of dimension $d_k$); and a value vector, $v$ (of dimension $d_v$). The so-called "hard attention" model returns the value that corresponds to the key that matches the query. A commonly used softer version of this is the scaled dot-product attention, which weights the $v$ vector by a softmax function applied to the scaled dot product of the $q$ and $k$ vectors. It is convenient to process multiple queries simultaneously which are packed together into a matrix $Q$ and apply them to similarly packed key and value matrices $K$ and $V$. Then, the whole process is governed by Eq. - 1.

$$\text{self-attention}(Q, K, V) = softmax\left(\frac{QK^T}{\sqrt{d_k}}\right)V \tag{1}$$

Here, the attention-weights are represented as $W = softmax(\frac{QK^T}{\sqrt{d_k}})$ and indicate how the components in the query correlate with different parts of the key. The scaling factor $1/\sqrt{d_k}$ is used to prevent the input to the softmax function from reaching the saturated region for large values of $d_k$.

### 3.2 Diffusion models

Diffusion models, first introduced by Sohl-Dickstein et al. (2015), have emerged as a noteworthy category of generative models that have consistently achieved state-of-the-art performance across a diverse range of data modalities including image data (Dhariwal & Nichol, 2021; Ho et al., 2020; 2022a), speech data (Chen et al., 2021; Kong et al., 2020), and video data (Ho et al., 2022b; Yang et al., 2022).

Diffusion-based probabilistic models have two distinct processes: a forward process and a reverse process. In the forward process, noise is incrementally added to the original data at each step, culminating in a state of pure noise after a series of iterations. Conversely, the reverse process systematically eliminates noise at each

step, commencing from a state of pure noise and gradually constructing a distribution that represents the original data, following an identical iteration count as in the forward process Sohl-Dickstein et al. (2015); Ho et al. (2020).

A diffusion model involves approximating a data distribution $q(X_0)$ by learning a model distribution $p_\theta(X_0)$. Let $X_t$ for $t = \{1, \dots, T\}$ be the latent variables representing the noisy data at diffusion step $t$. These belong to the same sample space as the original data, $X_0$. The forward diffusion process, which is a Markov chain is defined by

$$q(X_{1:T}|X_0) = \prod_{t=1}^{T} q(X_t|X_{t-1})$$

$$q(X_t|X_{t-1}) = \mathcal{N}(\sqrt{1-\beta_t}X_{t-1}, \beta_t\mathbf{I}) \tag{2}$$

Here, $\beta_t$ represents the variance of the noise applied at each diffusion step $t$ of the forward process. Furthermore, $X_t$ has a closed form $X_t = \sqrt{\bar{\alpha}_t}X_0 + \sqrt{(1-\bar{\alpha}_t)}\epsilon$, where $\alpha_t = 1 - \beta_t$, $\bar{\alpha}_t = \prod_{i=1}^{t} \alpha_i$, and $\epsilon \sim \mathcal{N}(0, \mathbf{I})$. The reverse process is also a Markov chain that denoises $X_t$ to get $X_{t-1}$ using a denoising function $\epsilon_\theta$. After $T$ such iterations, we regenerate $X_0$. The reverse process is defined by:

$$p_\theta(X_{0:T}) = p(X_T) \prod_{t=1}^{T} p_\theta(X_{t-1}|X_t), \text{ where } X_T \sim \mathcal{N}(0, \mathbf{I}) \text{ and}$$

$$p_\theta(X_{t-1}|X_t) = \mathcal{N}(\mu_\theta(X_t, t), \sigma_\theta(X_t, t)\mathbf{I}) \tag{3}$$

Here, $p_\theta(X_{t-1}|X_t)$ is a learnable function. Following (Ho et al., 2020), we have:

$$\mu_\theta(X_t, t) = \frac{1}{\sqrt{\alpha_t}}\left(X_t - \frac{\beta_t}{\sqrt{1-\bar{\alpha}_t}}\epsilon_\theta(X_t, t)\right) \tag{4}$$

where $\sigma_\theta(X_t, t)$ is kept constant for each diffusion step $t$. In Eq. 4, the $\epsilon_\theta$ denoising function is trainable. Using the parameterization of $\mu_\theta(X_t, t)$ in Eq. 4, Ho et al. (2020) have shown that the reverse process can be trained by optimizing the following objective:

$$L = \min_\theta \mathbb{E}_{X \sim q(X_0), \epsilon \sim \mathcal{N}(0, \mathbf{I}), t} ||\epsilon - \epsilon_\theta(X_t, t)||_2^2 \tag{5}$$

The denoising function $\epsilon_\theta$ takes the noisy data at step t, $X_t$, and the diffusion step $t$ as inputs and computes the added noise $\epsilon$ introduced to the noisy input $X_{t-1}$ to get $X_t$ in the forward diffusion step. Once the training is completed, we can sample $X_0$ by following the expressions outlined in Eqs. (6) and (4).

### 3.3 Probabilistic imputation with diffusion models

We extend the definition of diffusion for the time-series imputation problem by making the reverse process condition on the observed value for the posterior estimation. Given a sample $X_0$, we condition on the observed values, $X_0^{co}$ and generate the imputation targets, $X_0^{ta}$. In the case of probabilistic imputation, we approximate the data distribution $q(X_0^{ta}|X_0^{co})$ with model distribution $p_\theta(X_0^{ta}|X_0^{co})$. To do that, we extend Eqs. 6 and 4 to account for the conditional input $X_0^{co}$. We now have,

$$p_\theta(X_{t-1}^{ta}|X_t^{ta}, X_0^{co}) = \mathcal{N}(\mu_\theta(X_t^{ta}, X_0^{co}, t), \sigma_\theta(X_t^{ta}, X_0^{co}, t)\mathbf{I}) \tag{6}$$

$$\mu_\theta(X_t^{ta}, X_0^{co}, t) = \frac{1}{\sqrt{\alpha_t}}\left(X_t^{ta} - \frac{\beta_t}{\sqrt{1-\bar{\alpha}_t}}\epsilon_\theta(X_t^{ta}, X_0^{co}, t)\right) \tag{7}$$

Here, $X_t^{ta}$ represents the noisy data at diffusion step $t$ in place of the imputation targets (missing values). And, $\mu_\theta(X_t^{ta}, X_0^{co}, t)$ represents the function determining the mean of $X_{t-1}^{ta}$ from the inputs $X_t^{ta}$, diffusion step $t$, and the conditional data $X_0^{co}$. Similarly, $\sigma_\theta(X_t^{ta}, X_0^{co}, t)$ is the function determining the variance of $X_{t-1}^{ta}$. The variance is assumed as constant for each time step as illustrated in Eq. - 8.

$$\sigma_\theta(X_t^{ta}, X_0^{co}, t) = \frac{1 - \bar{\alpha}_{t-1}}{1 - \bar{\alpha}_t}\beta_t \tag{8}$$

## 4 Self-attention-based Diffusion Model for Time-series Imputation

Our model, SADI, leverages the general framework for the conditional diffusion process provided in the work of Ho et al. (2020) for time-series imputation. Figure 2a provides an overview of the architecture that models the denoising function, $\epsilon_\theta(X_t^{ta}, X_0^{co}, t)$, as depicted in Eq. 7. The CSDI model also learns the same $\epsilon_\theta$ function, but has a different architecture that models the feature and time correlations with two distinct transformers in sequence. The first transformer isolates each time step into distinct samples when modeling feature dependencies. The second transformer similarly segments the features into separate samples when modeling temporal correlations (see Figure 2b). Unfortunately, this segregation of time and features overlooks correlations that occur jointly. In contrast, we adopt a more general approach in the form of *feature dependency encoder* (FDE) and *gated temporal attention block* (GTA) to capture the joint correlations, drawing some inspiration from the SAITS model, which outperforms CSDI in some random missing and partial blackout tasks Du et al. (2023). The FDE consists of two main components. The first is a 1-D convolution that captures information on the locality of a time-series. The second is a self-attention layer that focuses attention on the feature dimension. Together, these two components capture feature correlations that are time-aware. The GTA (Gated Temporal Attention) component is then responsible for capturing the temporal dependencies. Our two-stage imputation process was designed after the two DMSA (Diagonally Masked Self-attention) blocks in SAITS (Du et al., 2023), which combine the two intermediary imputations by learning weights based on attention maps and missingness information. While our work draws inspiration from the SAITS model, it diverges in significant ways, particularly with the incorporation of the FDE to capture the time-aware feature correlations and the inclusion of a generative diffusion component. In scenarios where there are partial blackouts, it can be challenging to accurately predict time-series data due to missing feature values and extended time gaps. This is especially true when relying solely on accessible feature values or long temporal intervals. Our research goes beyond existing models like SAITS, which aren't generative and may not perform optimally in these contexts. Instead, our model incorporates a generative dimension inspired by models like CSDI to address the problem of partial blackouts better.

We show a side-by-side comparison of SADI and CSDI in Figure 2. CSDI uses a back-to-back transformer approach to capture feature and time correlation information, which is then represented as vectors along the channel dimension. On the other hand, SADI employs a single channel with two explicit self-attention-based components to capture both feature and time correlations. Despite having both feature and temporal encoders in sequence, there is a difference in the way SADI and CSDI capture feature dependencies. SADI's feature dependency encoder is time-aware when capturing feature dependencies, while CSDI models feature correlations separately for each time step.

SADI has *three* main components: the **feature dependency encoder** (FDE), which models feature correlations across time; the **gated temporal attention** (GTA) block, which captures temporal dependencies across different features; and finally, a **two-stage imputation** process that enhances the quality of imputations. We know that the diffusion denoising function is used to predict the noise added by the forward process. Nevertheless, as shown by Eq. 7, it is evident that predicting noise is equivalent to predicting imputation. Henceforth, our discussion will revolve around predicting imputation for various components, although the actual computation of values is indirectly based on noise prediction. Figure 6 illustrates a more detailed view of our model architecture.

### 4.1 Learning feature dependencies

In scenarios where there is partial blackout, some features may be present at the same time step, and the feature dependency encoder (FDE) plays a crucial role in capturing the relationships between these features. By learning and modeling these feature dependencies, SADI is able to make more accurate imputations and better handle the partial blackout scenario. The FDE computes a new representation of the time-series that captures the joint time-series level relationships between the features in the original data. This self-attention mechanism with layer normalization and feed-forward network puts attention on the feature dimension to capture time-series-wise correlations.

Our approach deviates from the CSDI methodology by considering the time-series data of each feature as a unified sequence. Rather than isolating individual time steps into separate instances to capture feature

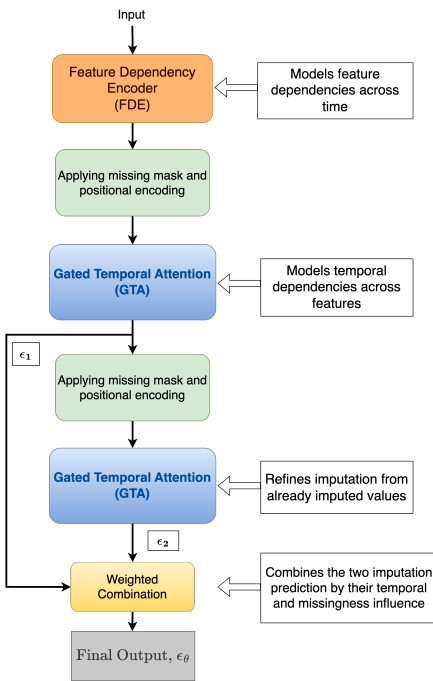
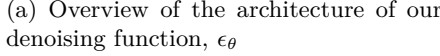
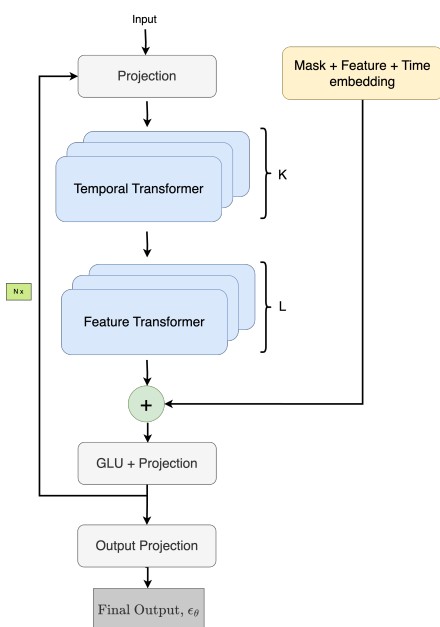

(a) Overview of the architecture of our denoising function, $\epsilon_\theta$

(b) Overview of the architecture of CSDI's $\epsilon_\theta$ function

Figure 2: A side-by-side comparison figure depicting the denoising function architectures of SADI and CSDI. CSDI utilizes a pair of consecutive transformers to capture both feature and temporal dependencies. In the context of modeling temporal relationships, they partition the features into $K$ distinct instances, with $K$ representing the number of features. Likewise, they segment time steps into $L$ separate instances, where $L$ corresponds to the length of the time series. On the other hand, SADI models feature and time dependencies jointly, thereby capturing the combined correlations of both features and time.

correlations, we utilize a 1-D convolution operation with a kernel size of $(1 \times 3)$ to extract locality information within the series. This approach uses a window of size 3 to iteratively update the value of the current time step, thus capturing the local temporal dynamics within the sequence. We then apply the attention mechanism to capture joint time-series level feature correlations. By placing the attention on the feature dimension after the convolution operation, we can effectively capture the dependencies between different features based on the time-series level relationships. This approach enables the FDE to better understand the complex relationships between different features in the time-series data.

FDE operates on both the observed values and the noisy data $(X_0^{co} + X_t^{ta})$ of dimensions $(L, K)$, subjecting them to the missingness mask, $M_0^{co}$ of the same dimensions. Furthermore, categorical positional encoding is applied to distinguish between different features within the feature dimension. A hyperparameter $N_{FDE}$ controls the number of FDE layers. The operations for FDE are shown in Eq. 9. Here, the output representation $\hat{X}$ has a dimensionality of $(L, K)$.

$$X = (feature\_pos\_enc(concat((X_0^{co} + X_t^{ta}), M_0^{co})))^T; \text{ X has dimensions } (K, L)$$

$$\hat{X} = \begin{cases} FDE_n(X) & \text{;if it is the } 1^{st} \text{ layer } (n = 1) \\ FDE_n(\hat{X}) & \text{;if } 1 < n \le N_{FDE} \end{cases} \qquad (9)$$

## 4.2 Learning temporal dependencies

To handle missing data in a partial blackout scenario, it is crucial for the model to take advantage of the available data on the temporal axis while making predictions. This requires the model to capture the relationship between different time steps across all features. Our model accomplishes this through the use of a gated temporal attention (GTA) block. This component specifically focuses on capturing temporal dependencies across different features, enabling the model to effectively leverage the available data to make accurate predictions, even in scenarios with significant data gaps.

The gated temporal attention (GTA) block resembles the residual block architecture within DiffWave (Kong et al. (2020)) and WaveNet (Oord et al. (2016)) models illustrated in Figure 6. Instead of employing dilated convolutions, our approach uses two self-attention layers focusing attention on the time dimension. The CSDI model also adopts the same residual block architecture, with its primary innovation being the inclusion of temporal and feature transformers within this block. In our case, the residual block, which we call GTA, only models temporal dependencies across features as the attention mechanism focuses on the time dimension. In this context, we introduce positional encoding in the time dimension to signify it as a sequence for the self-attention mechanism. There is a Gated Linear Unit (GLU) activation applied to the outputs of the last self-attention layer, hence, we named the block *gated temporal attention*.

The GTA block also has multiple layers controlled by the hyperparameter $N_{GTA}$. It operates on the output from the FDE, denoted as $\hat{X}_1$ of dimensions $(L, K)$ from Eq. 9 and $X_0^{co}$ $(L, K)$ after applying the missing mask $M_0^{co}$ and positional encoding on the time dimension. The positional encoding method is the sine-cosine encoding outlined in (Vaswani et al., 2017). We concatenate the missingness mask $M_0^{co}$ with both $\hat{X}_1$ and $X_0^{co}$ and project them to higher dimension with a feed-forward network (denoted as $linear()$ function in Eq. 10 and 11) and apply positional encoding to get $\hat{X}_{pos_1}$ $(L, D)$ and $X_{pos}^{co}$ $(L, D)$, where $D$ is the projected dimension. GTA takes $\hat{X}_{pos_1}$, $X_{pos}^{co}$, and the diffusion step embedding $t_{emb}$ as input and generates three outputs: a hidden state $\tilde{X}$ $(L, D)$ which is fed into the subsequent GTA layer as an input, a skip connection $\epsilon'$ $(L, D)$ contributing to the intermediary imputation, and attention weights $W_L$ $(L, L)$. To get the interim imputation $\epsilon_1$, we sum over all $\epsilon'$ $(L, D)$ skip connections and project them to dimensions $(L, K)$ according to Eq. 13. A detailed architecture of the GTA block can be found in Appendix A.2 Figure 6.

$$\hat{X}_{pos_1} = time\_pos\_enc(linear(concat(\hat{X}_1, M_0^{co}))) \tag{10}$$

$$X_{pos}^{co} = time\_pos\_enc(linear(concat(X_0^{co}, M_0^{co}))) \tag{11}$$

$$\tilde{X}, W_L, \epsilon' = \begin{cases} GTA_n^i(\hat{X}_{pos_i}, X_{pos}^{co}, t_{emb}) & \text{;if it is the } 1^{st} \text{ layer } (n = 1) \\ GTA_n^i(\tilde{X}, X_{pos}^{co}, t_{emb}) & \text{;if } 1 < n \leq N_{GTA} \end{cases} \tag{12}$$

$$\epsilon_1 = linear\left(\frac{\sum_n^{N_{GTA}} \epsilon_n'}{\sqrt{2}}\right) \tag{13}$$

## 4.3 Two-stage imputation process

After passing through multiple FDE and GTA layers, the data undergoes transformations that significantly alter its initial characteristics. This alteration is a part of the model's design to handle feature and temporal dependencies within the dataset. To deal with the potential loss of original data characteristics and improve imputation quality, we reintegrate the original noisy data into the process. This reintroduction serves as a grounding step, ensuring that the subsequent application of the *second GTA block* leverages both the transformed and original data characteristics for enhanced imputation results.

In the first stage, each GTA layer passes the hidden state $\tilde{X}$ to the subsequent GTA layer. However, these layers do not directly receive the imputation information, represented by $\epsilon'$, until the completion of the first

block, where they are aggregated into the initial interim imputation $\epsilon_1$. To enhance the prediction of missing values by leveraging already imputed data, $\epsilon_1$ is incorporated into the input for the second stage of GTA operations. The intuition behind this is that, initially, there are no imputed values to aid in the prediction process. But once the first set of imputations is generated, they are utilized to inform the prediction of the missing values in the subsequent block. This method not only leverages the imputed values themselves but also captures the relationships between observed and imputed data, as well as the impact imputed values have on other missing data points. The second GTA block carries out the same operations described in Section 4.2, yielding another interim imputation, denoted as $\epsilon_2$ and an attention weight matrix, $W_L$ of dimension $(L, L)$. Here, we introduce the original noisy data $X_t^{ta}$ into the input of the second GTA with the following operations to get the second interim imputation $\epsilon_2$.

$$\hat{X}_2 = \tilde{X} + \epsilon_1 + X_t^{ta}$$
$$\hat{X}_{pos_2} = time\_pos\_enc(linear(concat(\hat{X}_2, M_0^{co}))) \tag{14}$$

$$\tilde{X}, W_L, \epsilon' = \begin{cases} GTA_n^2(\hat{X}_{pos_2}, X_{pos}^{co}, t_{emb}) & \text{;if it is the } 1^{st} \text{ layer } (n = 1) \\ GTA_n^2(\tilde{X}, X_{pos}^{co}, t_{emb}) & \text{;if } 1 < n \leq N_{GTA} \end{cases} \tag{15}$$

$$\epsilon_2 = linear\left(\frac{\sum_n^{N_{GTA}} \epsilon'_n}{\sqrt{2}}\right) \tag{16}$$

Subsequently, we combine these two interim outputs, $\epsilon_1$ and $\epsilon_2$, to obtain the final imputation denoted as $\epsilon_\theta$ as depicted in Eq. 18. The weighted combination is designed to determine how much of each interim estimation should be used in order to produce the final estimation. These weighting coefficients, denoted as $\tilde{W}_L$ $(L, K)$, are acquired by applying missingness mask $M_0^{co}$ and a feed-forward network to project them to proper dimensions $(L, K)$ as shown in Eq. 17. In Eq. 18, $\odot$ is conventionally used to express element-wise product between two matrices/tensors.

$$\tilde{W}_L = sigmoid(linear(concat(W_L, M_0^{co}))) \tag{17}$$

$$\epsilon_\theta = (1 - \tilde{W}_L) \odot \epsilon_1 + \tilde{W}_L \odot \epsilon_2 \tag{18}$$

### 4.4 Training and sampling/inference

We adopt the training methodology outlined in Ho et al. (2020). This approach, illustrated in Algorithm 1, involves the uniform sampling of a diffusion step $t \in \{1, 2, \ldots, T\}$ during each training iteration in Step 2. Subsequently, it employs the forward process in closed form, denoted as $X_t^{ta} = \sqrt{\bar{\alpha}_t} X_0^{ta} + \sqrt{(1 - \bar{\alpha}_t)}\epsilon$, to progress to any time step $t$ of diffusion in a single step as illustrated in Step 5 of Algorithm 1. Our proposed denoising function, denoted as $\epsilon_\theta(X_t^{ta}, X_0^{co}, t)$, is designed to predict the noise component $\epsilon$ that must be removed from $X_t^{ta}$ during the reverse process in Step 6. To optimize our model, we formulate a loss function that minimizes the denoising loss in Step 7 as described in Eq. 5 for all three noise predictions produced by our denoising function: $\epsilon_1$, $\epsilon_2$, and $\epsilon_\theta$, as expressed by Eq. 19.

$$loss = \frac{M_0^{ta}}{2N}\left(||\epsilon - \epsilon_\theta||_2^2 + \frac{(||\epsilon - \epsilon_1||_2^2 + ||\epsilon - \epsilon_2||_2^2)}{2}\right) \tag{19}$$

In our experiments, we observed that focusing solely on optimizing the final $\epsilon_\theta$ prediction does not lead to the training loss converging to a favorable minima. However, when we optimize the loss function of all three predictions with respect to the ground truth, we achieve improved outcomes. Minimizing all three losses drives each of the predictions to the same ground truth. Ideally, all interim imputations should converge towards the same value and match the ground truth. Since the final prediction $\epsilon_\theta$ is a linear combination of

$\epsilon_1$ and $\epsilon_2$ and the loss function drives them to be the same value, ideally, the weights that combine the two interim predictions approach 0.5.

In Eq. 19, $N$ is the number of imputation targets and $M_0^{ta}$ is the target mask where 1 indicates the targets for imputation task and 0 represents observed values and original missing data (without ground truth). Let's assume $M$ is the mask where 0 represents the original missing values and $M_0^{co}$ is the mask where 0 represents both original and artificially created missing values (with ground truth). Then, the imputation target mask is $M_0^{ta} = M - M_0^{co}$.

---

**Algorithm 1** Training of our diffusion model

---

**Input:** Distribution of training data $X_0 \sim q(X_0)$, the number of iteration/epochs $N$, the list of noise levels $(\bar{\alpha}_1, \ldots, \bar{\alpha}_T)$

**Output:** Trained $\epsilon_\theta$ denoising function

1: **for** $i = 0$ to $N$ **do**
2:     $t \sim Uniform(\{1, \ldots, T\})$
3:     Separate $X_0$ into conditional observations $X_0^{co}$ and imputation targets $X_0^{ta}$
4:     Noise $\epsilon = \mathcal{N}(0, \mathbf{I})$ with the same dimension as $X_0^{ta}$
5:     One step calculation to noisy targets at step $t$, $X_t^{ta} = \sqrt{\bar{\alpha}_t} X_0^{ta} + \sqrt{(1 - \bar{\alpha}_t)}\epsilon$
6:     denoising function prediction, $\epsilon_1, \epsilon_2, \epsilon_\theta = \epsilon_\theta(X_t^{ta}, X_0^{co}, t)$
7:     Optimize the loss function for $\epsilon_\theta$, $\epsilon_1$, and $\epsilon_2$ according to Eq. (5) and (19).

---

During the inference phase shown in Algorithm 2, we generate imputed data for locations with missing values using a reverse diffusion process. This procedure is iterative, starting with pure Gaussian noise $X_T^{ta} \sim \mathcal{N}(0, \mathbf{I})$ at locations containing missing values. At each diffusion step $t$, we progressively remove some noise to produce the sample $X_{t-1}^{ta}$ for the preceding step, $t - 1$. To determine the noise to eliminate at diffusion step $t$, we utilize our proposed denoising function $\epsilon_\theta(X_t^{ta}, X_0^{co}, t)$ in Step 4. For producing $X_{t-1}^{ta}$, we first calculate the mean $\mu_\theta$ of $X_{t-1}^{ta}$ by removing the predicted noise $\epsilon_\theta$ from $X_t^{ta}$ in Step 5. The variance $\sigma_\theta$ in Step 6 remains constant for each diffusion step $t$ following the formulation from Ho et al. (2020). Step 7 to 10 show the formulation of $X_{t-1}^{ta}$ from the mean $\mu_\theta$ and the variance $\sigma_\theta$. Finally, we generate the output sample with predicted imputations in Step 12. We generate $N_{sample}$ such samples and use their mean as the final imputation.

---

**Algorithm 2** Sampling process

---

**Input:** Data sample $X_0$, missingness mask $M_0^{co}$, total number of diffusion steps $T$, trained denoising function $\epsilon_\theta$

**Output:** Imputed missing values $X_0^{ta}$

1: $X_0^{co} = $ observed values of $X_0$
2: $X_{curr} = X_T^{ta} \sim \mathcal{N}(0, \mathbf{I})$ (same dimensions as $X_0$)
3: **for** $t = T$ to 1 **do**
4:     $\epsilon_\theta = \epsilon_\theta(X_{curr}, X_0^{co}, M_0^{co})$
5:     $\mu_\theta = \frac{1}{\sqrt{\alpha_t}}(X_{curr} - \frac{\beta_t}{\sqrt{1-\bar{\alpha}_t}}\epsilon_\theta)$
6:     $\sigma_\theta = \frac{1-\bar{\alpha}_{t-1}}{1-\bar{\alpha}_t}\beta_t$; [Taken from Ho et al. (2020)]
7:     **if** $t = 0$ **then**
8:         $X_{curr} = \mathcal{N}(\mu_\theta, \mathbf{I})$
9:     **else**
10:         $X_{curr} = \mathcal{N}(\mu_\theta, \sigma_\theta \mathbf{I})$
11: $X_0^{ta} = X_{curr}$
12: $X_0 = X_0^{co} \times M_0^{co} + X_0^{ta} \times (1 - M_0^{co})$

---

## 5 Experiments

### 5.1 Experimental setup

We employed six synthetic datasets and four real-world datasets to assess the effectiveness of our model, SADI. We compared the performance of SADI with CSDI (Tashiro et al., 2021) (a conditional diffusion-based model), BRITS (Cao et al., 2018) (a bidirectional RNN-based autoregressive model), SAITS (Du et al., 2023) (a self-attention-based model), and MICE (van Buuren & Groothuis-Oudshoorn, 2011) (an iterative linear regression-based model) on both the synthetic datasets and the real-world datasets under partial blackout scenario. In every experimental trial, we randomly removed a number of blocks of data (2 in all our experiments), choosing a specified number of features uniformly from all features. For each absent block, we omitted a chosen number of consecutive time steps for the selected features. In the case of synthetic datasets, we eliminated 20 consecutive time steps. For the Air Quality dataset, we eliminated 10 consecutive time steps. For all other real-world datasets, we removed 30 consecutive time steps.

We trained the models once and tested them 20 times on the test set in different missingness settings (different ground truths) under partial blackout. For each test, we uniformly selected which features are missing and selected the number of blocks of such missingness (2 in all our experiments). As CSDI and SADI are generative models, we generate 50 predicted samples to approximate the probability distribution of the missing data. For SADI, we calculate the mean of these samples, while for CSDI, following the guidance from Tashiro et al. (2021), we take the median of the samples to determine the final prediction. The remaining three models make point predictions for imputation.

To assess the performance of SADI, we rely on two key metrics: Mean Squared Error (MSE) and the Continuous Ranked Probability Score (CRPS), introduced in Matheson & Winkler (1976), along with the inclusion of a 95% confidence interval. CRPS is a statistical metric used to measure the accuracy of probabilistic forecasts, especially in fields like meteorology. This metric measures the difference between the cumulative distribution function (CDF) of the predicted values and the CDF of the actual observed values. More precisely, CRPS is determined by calculating the integral of the squared differences between these two CDFs across all possible values. If, say, $F$ is a function that predicts a distribution and the ground truth is $y$, then the CRPS formulation is given in Eq. 20. A lower CRPS value indicates a more accurate prediction. Since CRPS is a measure of performance for generative models, we use this metric only for CSDI and SADI.

$$CRPS(F, y) = \int (F(x) - \mathbf{1}_{x \geq y})^2 dx \tag{20}$$

### 5.2 Synthetic time-series data

The goal of the synthetic datasets is to systematically study the strengths and weaknesses of our model as a function of the causal relationships in the data. We created six synthetic datasets denoted as v1, v2, v3, v4, v5, and v6. Figure - 3 illustrates the time-series features and their interrelationships in these synthetic datasets. Each dataset has a time-series length of 100. The circular nodes in Figure - 3 represent the features in each dataset, and the arrows depict the functional dependencies between these features. The solid arrows indicate dependencies between features within the same time step, while the dashed arrows indicate dependencies with the previous time step. In Figure 3, $\theta_t^i$ represents the input of the periodic functions (e.g. $sin(\theta_t^i)$) for feature $f_i$ at time $t$. All feature functions are constrained within a finite range. The characteristics of each dataset are detailed in Table 6. The independent features are generated by randomly selecting the lower and upper bound of the input within the specified limits outlined in Table 6, and then doing a linear interpolating from the lower bound to the upper bound to create the inputs to the periodic function for each time step $t$ for a given sample.

Tables - 1 and 2 show the comparative analysis of the performance of SADI against the other models using the six synthetic datasets. The first columns of the tables denote the dataset names, the second columns indicate the number of missing data blocks, and the third columns give the number of features with missing data in each experiment. The remaining columns show the mean squared error in Table 1 and CRPS in Table 2. Both tables show that with the exception of the dataset $v1$, SADI outperforms other models across

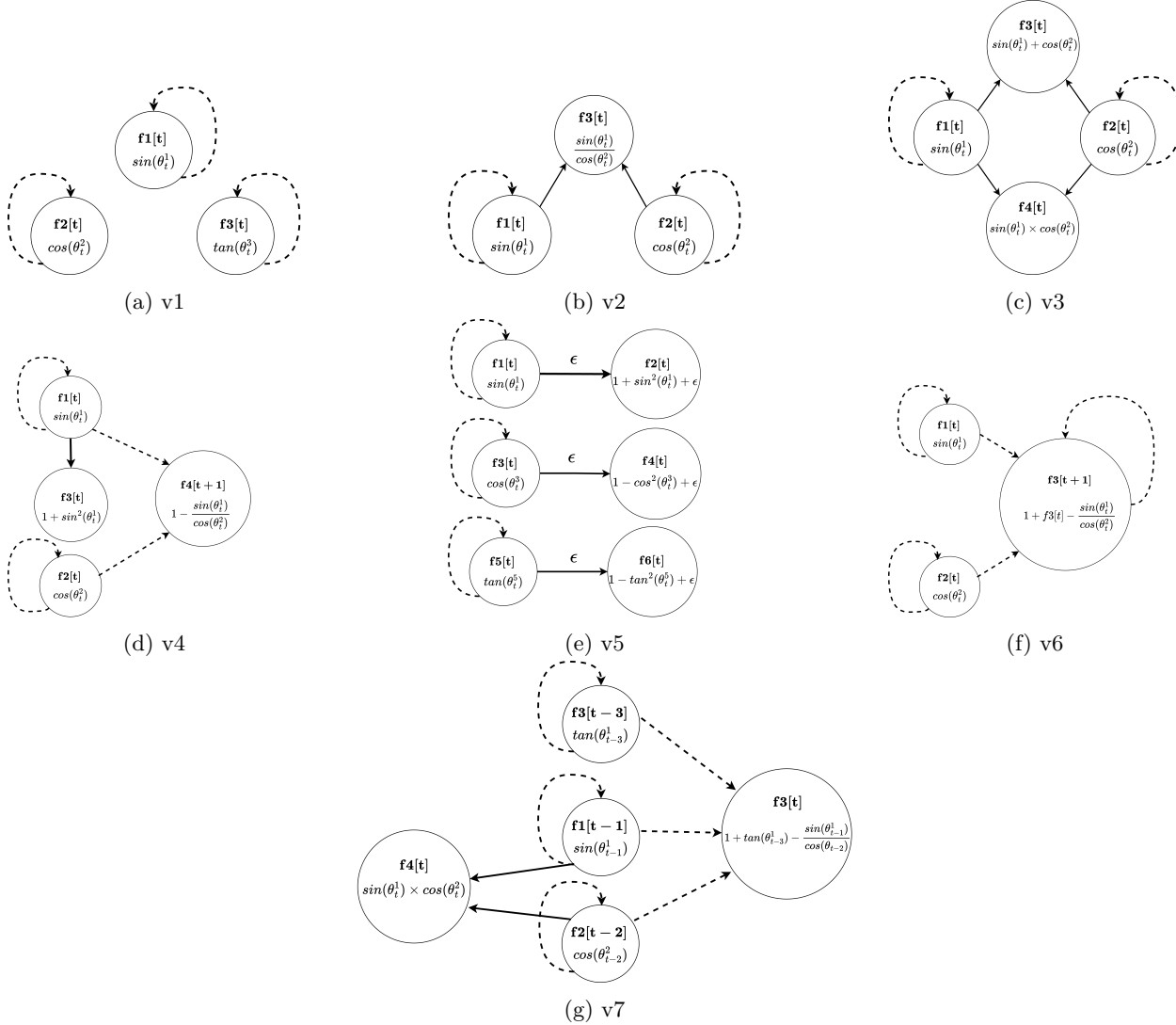

(a) v1  (b) v2  (c) v3

(d) v4  (e) v5  (f) v6

(g) v7

Figure 3: Synthetic datasets. The solid arrows indicate feature interconnections within the same time step, while the dashed arrows signify that the $t$-th time step's feature depends on the same or other features' $t-1$-th or $t-2$-th time step's value.

all synthetic experiments. For dataset $v1$, CSDI performs better than SADI in both MSE and CRPS, a circumstance that can be attributed to the absence of high feature correlations. This is corroborated by the data in Table 9, indicating that $v1$ possesses the lowest average Variance Inflation Factor (VIF) among its features. In statistical terms, a VIF score exceeding 5 signifies a high degree of multicollinearity or significant feature correlation, a situation that characterizes the other synthetic datasets. Consequently, SADI performs better when the dataset comprises features with high intercorrelation and fares worse than CSDI when the features exhibit limited correlation. As we will see from the ablation studies in Section 5.4, this is directly attributable to the FDE component of SADI, which is specifically designed to capture the dependencies between different features. In the case of $v1$, the FDE component tries to find dependencies where there is little correlation among the features, consequently hurting the performance. Dataset $v7$ has a non-markovian relationship among the features, and SADI can handle these type of relationships, too, as evident from Table - 1 and 2.

We conducted experiments with three additional synthetic datasets to support our claim that SADI performs better when there are more dependent features. These datasets are v8, v9, and v10, as illustrated in

| Datasets | # of blocks | # of Missing Features | MICE | BRITS | SAITS | CSDI | SADI |
|---|---|---|---|---|---|---|---|
| v1 | 2 | 1 | 0.02253 ± 0.00142 | 0.01253 ± 0.00085 | 0.00322 ± 0.00069 | **0.00150 ± 0.00082** | 0.00237 ± 0.00126 |
| | | 2 | 0.02268 ± 0.00088 | 0.01332 ± 0.00096 | 0.00291 ± 0.00046 | **0.00107 ± 0.00040** | 0.00159 ± 0.00087 |
| | | 3 | 0.02211 ± 0.00089 | 0.02031 ± 0.00196 | 0.00284 ± 0.00032 | **0.00174 ± 0.00040** | 0.00224 ± 0.00083 |
| v2 | 2 | 1 | 0.00454 ± 0.00049 | 0.00324 ± 0.00028 | 0.00033 ± 0.00007 | 0.00010 ± 0.00002 | **0.00005 ± 0.00002** |
| | | 2 | 0.01109 ± 0.00118 | 0.00396 ± 0.00024 | 0.00023 ± 0.00005 | 0.00012 ± 0.00003 | **0.00006 ± 0.00002** |
| | | 3 | 0.01305 ± 0.00094 | 0.00451 ± 0.00031 | 0.00020 ± 0.00004 | 0.00012 ± 0.00002 | **0.00005 ± 0.00001** |
| v3 | 2 | 1 | 0.00337 ± 0.00058 | 0.00571 ± 0.00073 | 0.00121 ± 0.00056 | 0.00013 ± 0.00005 | **0.00008 ± 0.00003** |
| | | 2 | 0.01051 ± 0.00172 | 0.00568 ± 0.00059 | 0.00189 ± 0.00042 | 0.00020 ± 0.00006 | **0.00012 ± 0.00005** |
| | | 3 | 0.01815 ± 0.00202 | 0.00604 ± 0.00032 | 0.00212 ± 0.00068 | 0.00022 ± 0.00009 | **0.00014 ± 0.00004** |
| | | 4 | 0.01983 ± 0.00165 | 0.00675 ± 0.00045 | 0.00285 ± 0.00054 | 0.00023 ± 0.00009 | **0.00021 ± 0.00007** |
| v4 | 2 | 1 | 0.00268 ± 0.00020 | 0.00586 ± 0.00060 | 0.00073 ± 0.00009 | 0.00008 ± 0.00003 | **0.00007 ± 0.00002** |
| | | 2 | 0.00831 ± 0.00152 | 0.00417 ± 0.00027 | 0.00052 ± 0.00014 | 0.00007 ± 0.00003 | **0.00005 ± 0.00001** |
| | | 3 | 0.01103 ± 0.00080 | 0.00482 ± 0.00044 | 0.00060 ± 0.00013 | 0.00008 ± 0.00002 | **0.00007 ± 0.00002** |
| | | 4 | 0.01451 ± 0.00104 | 0.00627 ± 0.00042 | 0.00059 ± 0.00007 | 0.00011 ± 0.00002 | **0.00009 ± 0.00002** |
| v5 | 2 | 1 | 0.0016 6± 0.00021 | 0.00371 ± 0.00036 | 0.00037 ± 0.00016 | 0.00028 ± 0.00007 | **0.00007 ± 0.00004** |
| | | 2 | 0.00989 ± 0.00244 | 0.00612 ± 0.00081 | 0.00036 ± 0.00013 | 0.00016 ± 0.00005 | **0.00009 ± 0.00005** |
| | | 3 | 0.01598 ± 0.00188 | 0.00722 ± 0.00062 | 0.00073 ± 0.00021 | 0.00017 ± 0.00006 | **0.00015 ± 0.00007** |
| | | 4 | 0.01882 ± 0.00233 | 0.00743 ± 0.00041 | 0.00054 ± 0.00015 | 0.00021 ± 0.00006 | **0.00017 ± 0.00007** |
| | | 5 | 0.02078 ± 0.00228 | 0.00898 ± 0.00074 | 0.00057 ± 0.00014 | 0.00017 ± 0.00005 | **0.00015 ± 0.00006** |
| v6 | 2 | 1 | 0.01628 ± 0.00175 | 0.00650 ± 0.00047 | 0.00048 ± 0.00016 | 0.00003 ± 0.00001 | **0.00002 ± 0.00001** |
| | | 2 | 0.01743 ± 0.00128 | 0.00698 ± 0.00047 | 0.00063 ± 0.00011 | 0.00005 ± 0.00002 | **0.00004 ± 0.00002** |
| | | 3 | 0.01948 ± 0.00111 | 0.00675 ± 0.00034 | 0.00062 ± 0.00013 | 0.00005 ± 0.00001 | **0.00004 ± 0.00001** |
| v7 | 2 | 1 | 0.00870 ± 0.00142 | 0.00585 ± 0.00130 | 0.00052 ± 0.00032 | 0.00022 ± 0.00022 | **0.00009 ± 0.00007** |
| | | 2 | 0.01167 ± 0.00155 | 0.00831 ± 0.00118 | 0.00040 ± 0.00013 | 0.00019 ± 0.00008 | **0.00010 ± 0.00006** |
| | | 3 | 0.01476 ± 0.00160 | 0.00951 ± 0.00159 | 0.00052 ± 0.00027 | 0.00026 ± 0.00013 | **0.00013 ± 0.00006** |

Table 1: Partial blackout scenario for **synthetic** datasets: MSE (with 95% confidence interval) is calculated by averaging 20 inference trials (length of missing time period = 20).

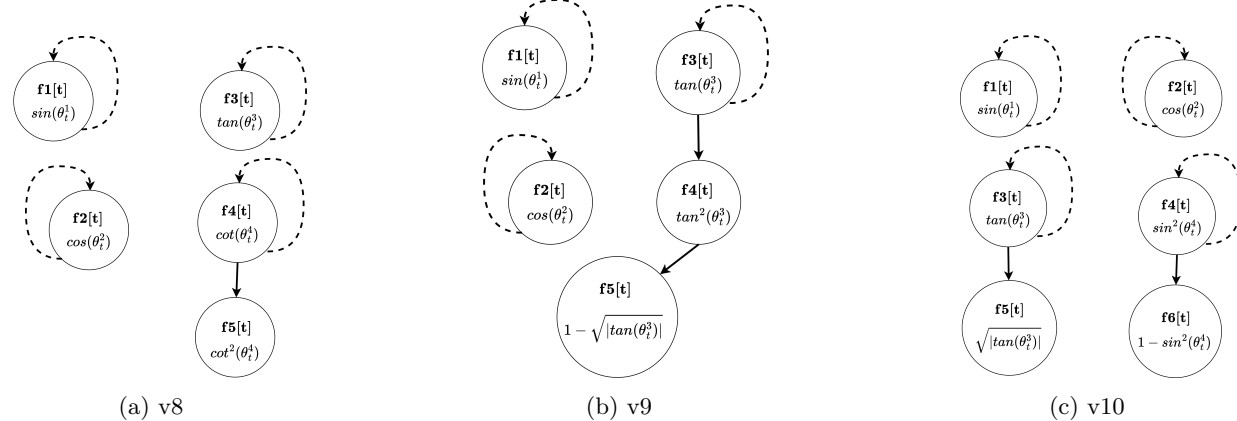

(a) v8  (b) v9  (c) v10

Figure 4: Synthetic datasets showing different levels of feature dependency. The dashed arrows represent dependency with the feature at the same time step.

Figures 4a, 4b, and 4c, respectively. Dataset v8 has three independent features and two interdependent features, while v9 has two independent features and three interdependent features. Dataset v10 has two independent features and two groups of two interdependent features. We present the results for these three datasets on the partial blackout scenario in Figure 5. The results indicate that for dataset v8, where there are more independent features than dependent ones, CSDI performs significantly better than SADI. However, for datasets v9 and v10, where there are more interdependent features than independent features, SADI outperforms CSDI by a huge margin.

| Datasets | # of blocks | # of Missing Features | CSDI | SADI |
|---|---|---|---|---|
| v1 | 2 | 1 | **0.00261 ± 0.00057** | 0.00352 ± 0.00094 |
| | | 2 | **0.00245 ± 0.00036** | 0.00307 ± 0.00060 |
| | | 3 | **0.00269 ± 0.00037** | 0.00339 ± 0.00056 |
| v2 | 2 | 1 | 0.00120 ± 0.00012 | **0.00081 ± 0.00010** |
| | | 2 | 0.00132 ± 0.00014 | **0.00085 ± 0.00011** |
| | | 3 | 0.00132 ± 0.00008 | **0.00079 ± 0.00007** |
| v3 | 2 | 1 | 0.00092 ± 0.00012 | **0.00090 ± 0.00013** |
| | | 2 | 0.00108 ± 0.00014 | **0.00107 ± 0.00012** |
| | | 3 | 0.00182 ± 0.00016 | **0.00114 ± 0.00015** |
| | | 4 | 0.00154 ± 0.00019 | **0.00133 ± 0.00019** |
| v4 | 2 | 1 | 0.00085 ± 0.00012 | **0.00083 ± 0.00013** |
| | | 2 | 0.00073 ± 0.00011 | **0.00070 ± 0.00009** |
| | | 3 | 0.00083 ± 0.00010 | **0.00080 ± 0.00010** |
| | | 4 | 0.00095 ± 0.00010 | **0.00094 ± 0.00010** |
| v5 | 2 | 1 | 0.00110 ± 0.00022 | **0.00082 ± 0.00012** |
| | | 2 | 0.00111 ± 0.00018 | **0.00091 ± 0.00019** |
| | | 3 | 0.00111 ± 0.00020 | **0.00110 ± 0.00018** |
| | | 4 | 0.00120 ± 0.00019 | **0.00119 ± 0.00017** |
| | | 5 | 0.00113 ± 0.00017 | **0.00109 ± 0.00017** |
| v6 | 2 | 1 | 0.00052 ± 0.00008 | **0.00044 ± 0.00007** |
| | | 2 | 0.00049 ± 0.00008 | **0.00040 ± 0.00008** |
| | | 3 | 0.00053 ± 0.00006 | **0.00046 ± 0.00007** |
| v7 | 2 | 1 | 0.00105 ± 0.00035 | **0.00076 ± 0.00023** |
| | | 2 | 0.00097 ± 0.00021 | **0.00075 ± 0.00020** |
| | | 3 | 0.00097 ± 0.00021 | **0.00076 ± 0.00015** |

Table 2: Partial blackout scenario for **synthetic** datasets: CRPS (with 95% confidence interval) is calculated by averaging 20 inference trials (length of missing time period = 20).

### 5.3 Real-world time-series data

The first real-world dataset we describe is a grape cultivar cold hardiness dataset from **AgAID**, which measures the resistance of the grape plants and their other characteristics along with environmental factors at regular points of time (Institue (2023)). The plant-related features were collected by the viticulture team from Washington State University[1]. The environmental data, on the other hand, was directly acquired through the AgWeatherNet API [2]. The dataset covers dormant seasons, commencing from September 7-th of one year and extending to May 15-th of the subsequent year. There are **21** features and **252** steps in time-series. It has a total of 34 seasons (1988 to 2022), where we used the first 32 seasons as the training data, and the last 2 as the test data.

**Air Quality** is a popular dataset considered in (Yi et al., 2016) among others. In accordance with prior research (Song et al., 2020b; Cao et al., 2018; Tashiro et al., 2021), we utilize hourly PM2.5 measurements from **36** stations (features) located in Beijing, covering a period of 12 months. We aggregate these measurements into time series, each consisting of **36** consecutive time steps. This dataset exhibits an approximate 13% rate of missing values. Importantly, the missing values follow some unknown patterns which are not completely random. This dataset includes a distinct test set that provides ground truth values for the missing data. We applied our customized missing scenarios to this test data.

---

[1]https://wine.wsu.edu/

[2]https://weather.wsu.edu/

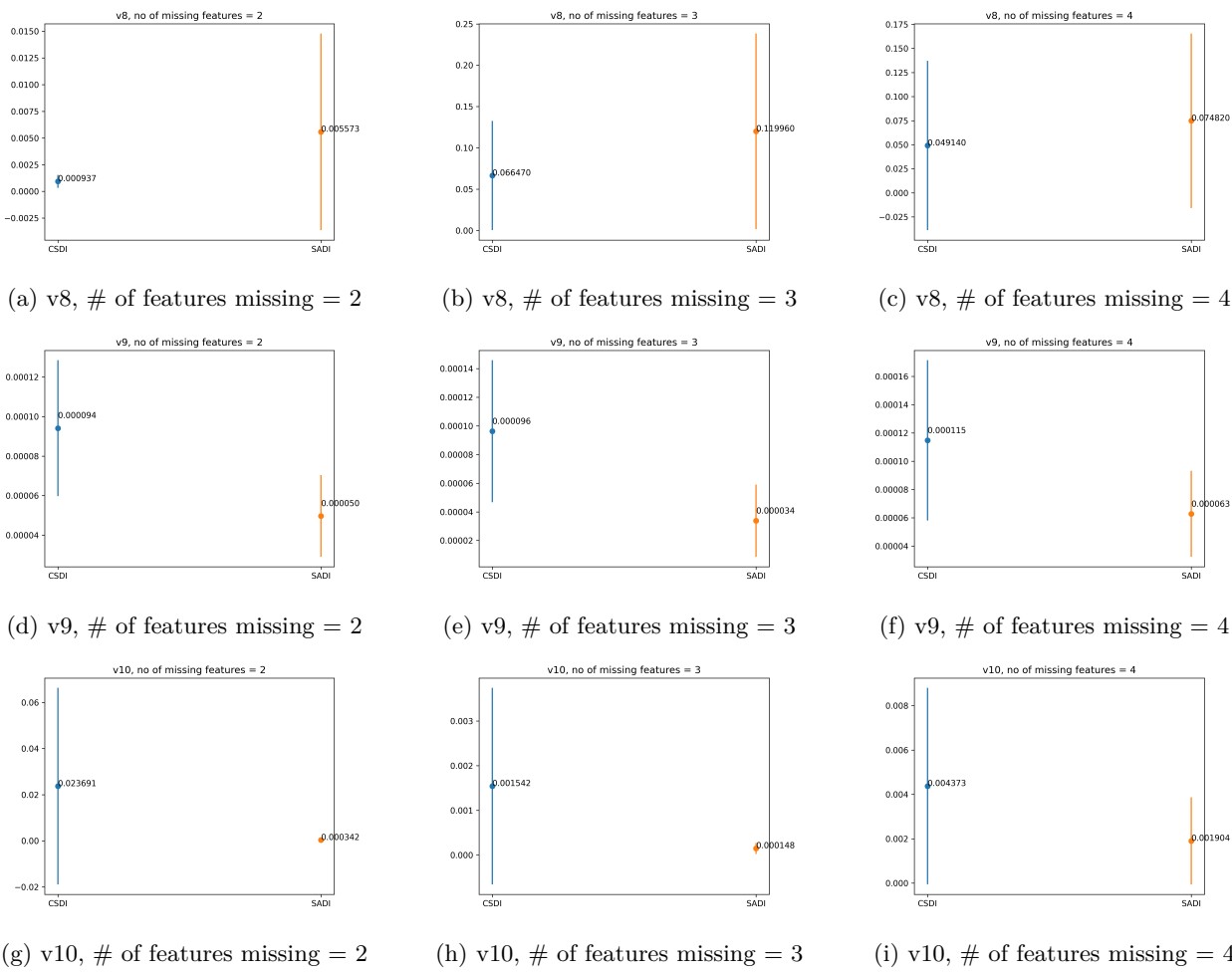

Figure 5: MSE ± 95% confidence interval for CSDI and SADI. The dot shows the MSE value, and the bar represents the 95% confidence interval. The length of missingness in the time series is 20 in these experiments.

Another widely known dataset is the **Electricity** Load Diagram from the public UCI machine learning repository Dua & Graff (2017). It comprises electricity consumption data, measured in kilowatt-hours (kWh), gathered from 370 clients at 15-minute intervals. This dataset has no original missing data. There are 48 months worth of data from January 1, 2011, to December 31, 2014. This dataset has **370** features and **100** time steps. Since this dataset has no missing data, we artificially removed 20% of the data when training our model on it. We designate the first 10 months of data (from January 2011 to October 2011) as the test set, the subsequent 10 months (from November 2011 to August 2012) as the validation set, and the remaining data (from September 2012 to December 2014) as the training set, the same as Du et al. (2023).

The last dataset examined in this study consists of temperature data sourced from the Northwest Alliance for Computational Science & Engineering ( **NACSE**) PRISM climate data PRISM (2014). It has the maximum and minimum temperatures recorded daily across 176 weather stations in Oregon. The dataset comprises **352** features, and the time-series spans a year, specifically **366** days. The dataset contains data for a total of 11 years, from January 1, 2011, to December 31, 2021. For our experimental setup, we designate the first 9 years as the training data and reserve the last 2 years for testing.

Tables 3 and 4 show the MSE and CRPS results with 95% confidence interval for the four real-world datasets. The experimental setup and the columns of the tables are the same as previously discussed. Here, we can observe that SADI outperforms the other models in MSE and CRPS in partial blackout scenarios. Our model, SADI, demonstrates superior performance compared to other models, even in instances involving random

| Datasets | # of blocks | # of Missing Features | MICE | BRITS | SAITS | CSDI | SADI |
|---|---|---|---|---|---|---|---|
| AgAID | 2 | 1 | $0.0050 \pm 0.0028$ | $0.0087 \pm 0.0048$ | $0.0016 \pm 0.0008$ | $0.0026 \pm 0.0021$ | $\mathbf{8.6e\text{-}05 \pm 9.7e\text{-}05}$ |
| | | 3 | $0.0156 \pm 0.0072$ | $0.0135 \pm 0.0068$ | $0.0022 \pm 0.0010$ | $0.0045 \pm 0.0019$ | $\mathbf{0.0004 \pm 0.0003}$ |
| | | 5 | $0.0205 \pm 0.0065$ | $0.0137 \pm 0.0064$ | $0.0027 \pm 0.0014$ | $0.0058 \pm 0.0026$ | $\mathbf{0.0004 \pm 0.0003}$ |
| | | 7 | $0.0252 \pm 0.0051$ | $0.0150 \pm 0.0031$ | $0.0036 \pm 0.0012$ | $0.0074 \pm 0.0051$ | $\mathbf{0.0005 \pm 0.0005}$ |
| | | 9 | $0.0459 \pm 0.0126$ | $0.0256 \pm 0.0100$ | $0.0045 \pm 0.0013$ | $0.0049 \pm 0.0017$ | $\mathbf{0.0003 \pm 0.0002}$ |
| | | 11 | $0.0582 \pm 0.0159$ | $0.0335 \pm 0.0113$ | $0.0055 \pm 0.0015$ | $0.0059 \pm 0.0021$ | $\mathbf{0.0006 \pm 0.0004}$ |
| Air Quality | 2 | 1 | $0.0280 \pm 0.0100$ | $0.0281 \pm 0.0101$ | $0.0280 \pm 0.0101$ | $0.00223 \pm 0.00244$ | $\mathbf{0.00174 \pm 0.00191}$ |
| | | 3 | $0.02074 \pm 0.0183$ | $0.0200 \pm 0.0188$ | $0.0206 \pm 0.0183$ | $0.00185 \pm 0.00129$ | $\mathbf{0.00104 \pm 0.0004}$ |
| | | 5 | $0.0166 \pm 0.0072$ | $0.0174 \pm 0.0073$ | $0.0166 \pm 0.0071$ | $0.00153 \pm 0.00063$ | $\mathbf{0.00109 \pm 0.00059}$ |
| | | 7 | $0.0223 \pm 0.0127$ | $0.0220 \pm 0.0127$ | $0.0225 \pm 0.0128$ | $0.00154 \pm 0.00040$ | $\mathbf{0.00130 \pm 0.00065}$ |
| | | 9 | $0.0214 \pm 0.0160$ | $0.0203 \pm 0.0151$ | $0.0203 \pm 0.0153$ | $0.00147 \pm 0.00059$ | $\mathbf{0.00106 \pm 0.00032}$ |
| | | 11 | $0.0197 \pm 0.0066$ | $0.0192 \pm 0.0062$ | $0.0192 \pm 0.0065$ | $0.00124 \pm 0.00021$ | $\mathbf{0.00107 \pm 0.00020}$ |
| Electricity | 2 | 1 | $0.5466 \pm 0.1231$ | $0.8743 \pm 0.4098$ | $0.8312 \pm 0.2567$ | $0.3569 \pm 0.0498$ | $\mathbf{0.1726 \pm 0.0892}$ |
| | | 10 | $0.5991 \pm 0.0303$ | $1.0313 \pm 0.0474$ | $0.9915 \pm 0.0621$ | $0.6317 \pm 0.0590$ | $\mathbf{0.1379 \pm 0.0125}$ |
| | | 15 | $0.5790 \pm 0.0199$ | $0.9393 \pm 0.0770$ | $0.9003 \pm 0.0394$ | $0.5703 \pm 0.0832$ | $\mathbf{0.1273 \pm 0.0191}$ |
| | | 20 | $0.5757 \pm 0.0199$ | $0.9239 \pm 0.0429$ | $0.8877 \pm 0.0252$ | $0.5863 \pm 0.0477$ | $\mathbf{0.1346 \pm 0.0478}$ |
| | | 30 | $0.6027 \pm 0.0100$ | $0.9979 \pm 0.0134$ | $0.9387 \pm 0.0169$ | $0.56090 \pm 0.03535$ | $\mathbf{0.1248 \pm 0.0066}$ |
| | | 100 | $0.6916 \pm 0.0289$ | $1.009 \pm 0.0202$ | $0.9366 \pm 0.0239$ | $0.4395 \pm 0.0209$ | $\mathbf{0.1346 \pm 0.0072}$ |
| NACSE | 2 | 2 | $\mathbf{0.0044 \pm 0.0011}$ | $0.0071 \pm 0.0030$ | $0.0091 \pm 0.0056$ | $0.0205 \pm 0.0032$ | $0.0045 \pm 0.0038$ |
| | | 10 | $0.0051 \pm 0.0008$ | $0.0082 \pm 0.0021$ | $0.01092 \pm 0.0027$ | $0.0191 \pm 0.0016$ | $\mathbf{0.0049 \pm 0.0018}$ |
| | | 50 | $0.0070 \pm 0.0009$ | $0.0083 \pm 0.0009$ | $0.0100 \pm 0.0013$ | $0.0191 \pm 0.0011$ | $\mathbf{0.0064 \pm 0.0008}$ |
| | | 90 | $0.0116 \pm 0.0008$ | $0.0117 \pm 0.0010$ | $0.0109 \pm 0.0010$ | $0.0210 \pm 0.0011$ | $\mathbf{0.0071 \pm 0.0008}$ |
| | | 100 | $0.0117 \pm 0.0012$ | $0.0117 \pm 0.0013$ | $0.0106 \pm 0.0011$ | $0.0206 \pm 0.0012$ | $\mathbf{0.0073 \pm 0.0007}$ |

Table 3: Partial blackout scenario: MSE (with 95% confidence interval) is calculated by averaging 20 inference trials (length of missing time period = 10 (Air Quality) and 30 (for the rest)).

missing data, complete blackout, and forecasting as illustrated in Appendix A.5. In these experiments, we have observed that CSDI (code taken from one of the author's GitHub reporsitory[3])requires a huge amount of GPU memory when dealing with high dimensional data such as - Electricity and NACSE datasets. For these two datasets, we had to reduce the number of channels to 4 because of our GPU constraints, which may have had some negative effect on its performance shown in Tables 3 and 4.

### 5.4 Ablation Study

Our model, SADI, has three core components: (1) the FDE (feature dependency encoder) block that models feature inter-correlations, (2) the two-stage imputation process, and (3) the weighted combination of the two intermediate imputations. Now, we will do an ablation study to show the impact of these three design decisions. The models for ablation are -

- **SADI**: The SADI model with all of its components.

- **No FDE**: SADI model without the FDE component.

- **No 2nd block**: SADI model after removing the second stage of imputation. Instead of having two separate $N_{GTA}$ layers for each block, we now have a single block with $2 \times N_{GTA}$ layers. It takes the first stage's output as the final imputation.

- **No wt. comb.**: SADI model without the weighted combination of two blocks. It takes the prediction of the second stage as the final imputation.

Table - 5 shows the ablation results for all three design choices. The ablation study was done on six synthetic datasets, the AgAID dataset, and the NACSE dataset. For every model, the training was done 3 times and each time there were 20 inference runs with different missing scenarios. Table 5 reports the MSE with 95% confidence interval averaging across all training and inference runs. Note that all values in Table 5 should

---

[3]https://github.com/ermongroup/CSDI

| Datasets | # of blocks | # of Missing Features | CSDI | SADI |
|----------|-------------|------------------------|------|------|
| AgAID | 2 | 1 | 0.0089 ± 0.0037 | **0.0033 ± 0.0038** |
| | | 3 | 0.0092 ± 0.0040 | **0.0022 ± 0.0009** |
| | | 5 | 0.0085 ± 0.0029 | **0.0023 ± 0.0009** |
| | | 7 | 0.0087 ± 0.0017 | **0.0024 ± 0.0004** |
| | | 9 | 0.0093 ± 0.0020 | **0.0024 ± 0.0006** |
| | | 11 | 0.0095 ± 0.0018 | **0.0027 ± 0.0005** |
| Air Quality | 2 | 1 | 0.0094 ± 0.0079 | **0.0073 ± 0.0062** |
| | | 3 | 0.0047 ± 0.0010 | **0.0039 ± 0.0007** |
| | | 5 | 0.0042 ± 0.0004 | **0.0034 ± 0.0004** |
| | | 7 | 0.0039 ± 0.0014 | **0.0028 ± 0.0009** |
| | | 9 | 0.0051 ± 0.0013 | **0.0041 ± 0.0011** |
| | | 11 | 0.0050 ± 0.0012 | **0.0041 ± 0.0012** |
| Electricity | 2 | 1 | 0.1383 ± 0.0202 | **0.0556 ± 0.0011** |
| | | 10 | 0.1488 ± 0.0059 | **0.0515 ± 0.0016** |
| | | 15 | 0.1477 ± 0.0040 | **0.0486 ± 0.0020** |
| | | 20 | 0.1517 ± 0.0023 | **0.0488 ± 0.0020** |
| | | 30 | 0.1518 ± 0.0027 | **0.0480 ± 0.0025** |
| | | 100 | 0.1557 ± 0.0030 | **0.0566 ± 0.0035** |
| NACSE | 2 | 2 | 0.0263 ± 0.0021 | **0.0165 ± 0.0030** |
| | | 10 | 0.0258 ± 0.0018 | **0.0155 ± 0.0010** |
| | | 50 | 0.0268 ± 0.0018 | **0.0151 ± 0.0010** |
| | | 90 | 0.0258 ± 0.0016 | **0.0146 ± 0.0009** |
| | | 100 | 0.0249 ± 0.0018 | **0.0137 ± 0.0008** |

Table 4: Partial blackout scenario: CRPS (with 95% confidence interval) is calculated by averaging 20 inference trials (length of missing time period = 10 (Air Quality) and 30 (for the rest)).

be multiplied by $10^{-3}$ for true MSE and confidence interval. From Table 5, we observe that with all of its components, SADI outperforms the versions that lack at least one of the core components in all but one dataset. The only exceptuon is dataset v1, where the model **No FDE** outperforms SADI. This outcome aligns with our expectations, given that SADI's FDE component tries to identify correlations in situations where such associations are scarce or nonexistent.

It can be observed that for most of the datasets, removing the second stage leads to a significant decrease in performance. This is particularly true for the two synthetic datasets that contain features with dependencies on the previous time step values of other features. The difference in performance is highly visible in the two real-world datasets. The difference is not much in the case of synthetic datasets because they are composed of fairly simple periodic functions, and the models can predict them well even if some components are missing. Finally, the absence of weighted combination damages performance significantly in the two real-world datasets and the 3 synthetic datasets v2, v3, and v5. These datasets have high correlation among features of the same time step. This component matters only slightly in case of the datasets whose features have interdependency with previous time step's features as observed from the results of v4 and v6 in Table 5.

## 6 Discussion and future work

In this paper, we address the multivariate time-series imputation, which is a critical problem in many domains. While existing research in time series imputation has made significant progress, a common limitation has been the concentration on a narrow set of cases of missingness, leaving a gap in understanding and addressing more realistic missing data scenarios. To close this gap, we introduce a flexible framework called **partial blackouts** that includes a wide variety of missing data patterns found in the real world. This notion provides a more thorough and practical evaluation of imputation models. Our approach to training and evaluation hinges on the MCAR (missing completely at random) strategy, as the missingness masks are independent of any observed data. Within the partial blackout framework, there exists a degree of correlation in missing values, yet it does not qualify as MAR (missing at random) because the absence of values

| Datasets | # of blocks | # of Missing Features | MSE $\times 10^3$ $\pm$ 95% CI $\times 10^3$ | | | |
|---|---|---|---|---|---|---|
| | | | SADI | No FDE | No 2nd block | No wt. comb. |
| v1 | 2 | 1 | $2.3 \pm 1.2$ | $\mathbf{2.172 \pm 0.55}$ | $2.2 \pm 0.65$ | $2.668 \pm 0.733$ |
| | | 2 | $1.5 \pm 0.08$ | $\mathbf{1.386 \pm 0.515}$ | $1.52 \pm 1.3$ | $2.122 \pm 0.373$ |
| | | 3 | $2.2 \pm 0.08$ | $\mathbf{1.668 \pm 0.210}$ | $1.97 \pm 0.72$ | $1.703 \pm 0.248$ |
| v2 | 2 | 1 | $\mathbf{0.064 \pm 0.01}$ | $0.074 \pm 0.011$ | $0.861 \pm 0.805$ | $0.069 \pm 0.017$ |
| | | 2 | $\mathbf{0.053 \pm 0.021}$ | $0.137 \pm 0.021$ | $3.419 \pm 2.531$ | $0.144 \pm 0.017$ |
| | | 3 | $\mathbf{0.061 \pm 0.022}$ | $0.185 \pm 0.021$ | $21.34 \pm 7.075$ | $0.199 \pm 0.016$ |
| v3 | 2 | 1 | $\mathbf{0.083 \pm 0.033}$ | $0.086 \pm 0.021$ | $0.238 \pm 0.111$ | $0.087 \pm 0.027$ |
| | | 2 | $\mathbf{0.1 \pm 0.048}$ | $0.113 \pm 0.14$ | $0.486 \pm 0.201$ | $0.116 \pm 0.008$ |
| | | 3 | $\mathbf{0.1 \pm 0.04}$ | $0.128 \pm 0.017$ | $0.536 \pm 0.152$ | $0.121 \pm 0.015$ |
| | | 4 | $\mathbf{0.2 \pm 0.071}$ | $0.207 \pm 0.02$ | $0.483 \pm 0.139$ | $0.265 \pm 0.041$ |
| v4 | 2 | 1 | $\mathbf{0.077 \pm 0.026}$ | $0.085 \pm 0.015$ | $10.15 \pm 3.997$ | $0.079 \pm 0.010$ |
| | | 2 | $\mathbf{0.052 \pm 0.014}$ | $0.108 \pm 0.017$ | $28.83 \pm 14.98$ | $0.088 \pm 0.013$ |
| | | 3 | $\mathbf{0.074 \pm 0.019}$ | $0.127 \pm 0.018$ | $35.45 \pm 5.916$ | $0.102 \pm 0.013$ |
| | | 4 | $\mathbf{0.094 \pm 0.019}$ | $0.125 \pm 0.011$ | $47.71 \pm 10.31$ | $0.121 \pm 0.019$ |
| v5 | 2 | 1 | $0.078 \pm 0.035$ | $0.108 \pm 0.062$ | $4.199 \pm 0.884$ | $\mathbf{0.058 \pm 0.041}$ |
| | | 2 | $\mathbf{0.091 \pm 0.058}$ | $0.098 \pm 0.048$ | $4.732 \pm 0.760$ | $0.129 \pm 0.049$ |
| | | 3 | $\mathbf{0.015 \pm 0.075}$ | $0.105 \pm 0.042$ | $4.111 \pm 0.701$ | $0.175 \pm 0.067$ |
| | | 4 | $\mathbf{0.170 \pm 0.069}$ | $0.182 \pm 0.068$ | $4.768 \pm 0.984$ | $0.183 \pm 0.049$ |
| | | 5 | $\mathbf{0.150 \pm 0.060}$ | $0.169 \pm 0.053$ | $4.536 \pm 0.535$ | $0.161 \pm 0.056$ |
| v6 | 2 | 1 | $\mathbf{0.029 \pm 0.013}$ | $0.043 \pm 0.016$ | $16.89 \pm 12.66$ | $0.061 \pm 0.034$ |
| | | 2 | $\mathbf{0.040 \pm 0.016}$ | $0.044 \pm 0.020$ | $237.3 \pm 94.75$ | $0.043 \pm 0.016$ |
| | | 3 | $\mathbf{0.042 \pm 0.012}$ | $0.058 \pm 0.015$ | $265.4 \pm 56.62$ | $0.046 \pm 0.018$ |
| AgAID | 2 | 3 | $\mathbf{0.46 \pm 0.35}$ | $1.43 \pm 0.9$ | $4.059 \pm 4.339$ | $0.78 \pm 0.56$ |
| | | 7 | $\mathbf{0.59 \pm 0.55}$ | $1.77 \pm 1.72$ | $2.219 \pm 1.152$ | $5.41 \pm 9.4$ |
| | | 11 | $\mathbf{0.63 \pm 0.40}$ | $1.64 \pm 0.8$ | $6.516 \pm 5.201$ | $6.75 \pm 3.87$ |
| NACSE | 2 | 10 | $\mathbf{4.9 \pm 1.8}$ | $37.44 \pm 20.46$ | $59.5 \pm 18.28$ | $39.53 \pm 16.15$ |
| | | 50 | $\mathbf{6.4 \pm 0.8}$ | $78.19 \pm 14.94$ | $95.08 \pm 22.97$ | $64.83 \pm 18.11$ |
| | | 100 | $\mathbf{7.3 \pm 0.7}$ | $86.08 \pm 12.19$ | $80.25 \pm 15.77$ | $63.65 \pm 9.12$ |

Table 5: Ablation Study of the three core components of SADI: MSE (with 95% confidence interval) is calculated by averaging accross three training runs each running 20 inference trials.

does not depend on any observed ones. The missingness in MCAR is independent of any known or unknown values, while in MAR it depends on some observed values. This MCAR setup for both our evaluation and training is also consistent across prior research works.

Another common drawback in many existing works is the absence of explicit modeling of feature dependencies at the time-series level, leading to suboptimal imputation quality. Among the state-of-the-art models, only the CSDI model addresses capturing feature dependencies as a distinct consideration. Our proposed solution, SADI, explicitly models both feature and temporal correlations. This design choice provides a robust approach for imputing data in scenarios involving diverse feature dimensions and interrelationships. Furthermore, our two-step imputation procedure takes into account the impact that imputed values have on further imputations, aiming to enhance the overall quality of the imputation. On the other hand, the imputation generated in the second stage may not always be better than the initial imputation in terms of quality. To address this concern, we have developed a dynamic weighting mechanism between the outputs of the two stages where the weights are determined by learnable parameters. The ablation experiments show that this weighting mechanism is critical for the success of our method.

In Section 5, we have demonstrated that SADI exhibits better performance compared to the state-of-the-art. We also observe that, with the exception of the $v1$ dataset, SADI outperforms CSDI across all datasets. Notably, the $v1$ dataset has lower multicollinearity among its features, as indicated by the average Variance

Inflation Factor (VIF) values in Table 9. This table shows that all other datasets, excluding $v1$, have an average VIF exceeding 5, which is typically considered indicative of high feature correlation in statistics. SADI performs better in scenarios where the dataset has high multicollinearity among the features. Conversely, its performance does not surpass that of the state-of-the-art when multicollinearity is minimal or absent. This discrepancy arises from the fact that the FDE component of SADI attempts to model feature correlations, even in cases where none exist as evident by ablation study where we removed the FDE component (See Table 5, first and second columns).

Additionally, our model requires a lower amount of GPU memory compared to CSDI for both training and inference, which makes it a viable option for diverse applications. When executing CSDI on large datasets such as Electricity and NACSE, we had to decrease the hyperparameter related to the number of channels due to GPU memory constraints. In contrast, SADI did not necessitate any hyperparameter reductions to accommodate the same GPU memory capacity.

Our primary focus in this paper is on modeling the feature and time correlations in the setting of low-to-high dimensional feature space. While we investigate the performance of our model in the partial blackout scenario in high dimensional feature space, our focus remains attuned to the short-to-medium term time series length. Our study does not encompass capturing long-term dependencies in time series data. Consequently, in our evaluation, we have not included the SSSD (Alcaraz & Strodthoff, 2022) model, as it falls within the domain of capturing long-term time dependencies and cannot work with a large number of features. Instead, we have chosen to take the CSDI model as the representative of the diffusion-based models for the scope of our investigation.

The current diffusion framework our model uses is borrowed from DDPM (Ho et al., 2020), which shows a notable time delay in the sampling process. This delay might be minimized by adopting the newer diffusion frameworks like improved DDPM (Song & Ermon, 2020), DDIM (Song et al., 2020a) and Schrödinger Bridge (De Bortoli et al., 2021; Chen et al., 2023). Another limitation of our model becomes apparent in case of datasets having minimal multicollinearity among their features, where it underperforms. This limitation also offers opportunities for future improvement.

## 7 Conclusion

In summary, this paper introduces the "partial blackout" framework to address multivariate time-series imputation, providing a more realistic evaluation of imputation models. Our model, SADI, explicitly models feature and temporal correlations and performs well except in datasets with minimal feature correlations. We have an innovative two-stage imputation process that enhances the quality of imputation. It requires less GPU memory than existing state-of-the-art models like CSDI. However, SADI does not utilize spatial information in spatiotemporal data and the diffusion framework it adopts has a slow sampling process. Future work aims to overcome these limitations.

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

# A  Appendix

## A.1  Synthetic data generation

We create six artificial datasets containing basic periodic functions as features. The input for these functions is represented as $\theta^i$, where $i$ corresponds to the $fi$ feature of the dataset. Each $\theta^i$ is constrained within a specified range, as detailed in Table - 6. We perform linear interpolation between the lower and upper bounds over 100 time steps and apply the periodic functions accordingly, as depicted in Table - 6. The dataset construction follows the outlined procedure:

- **v1:** All features are independent of each other and each feature only depends on the previous time steps value of itself.

- **v2:** Features have some dependency to each other in the same time step.

- **v3:** Two features have some dependency and one has linear dependency and the other has non-linear dependency.

- **v4:** Has a feature which is dependent on the other features' values from the previous time step.

- **v5:** Groups of interdependent features with some noise added to them.

- **v6:** Has a feature dependent on the values other features and itself from the previous time step.

| Dataset name | Number of features | $\theta^i$ bounds | Features |
|---|---|---|---|
| v1 | 3 | $\theta^1 = [0.00001, \frac{2\pi}{3})$, $\theta^2 = [0, 2\pi)$, $\theta^3 = [-\frac{\pi}{3}, \frac{\pi}{3})$ | $f1[t] = sin(\theta_t^1),\ f2[t] = cos(\theta_t^2),\ f3[t] = tan(\theta_t^3)$ |
| v2 | 3 | $\theta^1 = [-\frac{\pi}{3}, \frac{\pi}{3})$, $\theta^2 = [-\frac{\pi}{4}, \frac{\pi}{3})$ | $f1[t] = sin(\theta_t^1),\ f2[t] = cos(\theta_t^2),\ f3[t] = \frac{f1[t]}{f2[t]}$ |
| v3 | 4 | $\theta^1 = [0.0001, \frac{2\pi}{3})$, $\theta^2 = [0, \pi)$ | $f1[t] = sin(\theta_t^1),\ f2[t] = cos(\theta_t^2)$, $f3[t] = f1[t] + f2[t],\ f4[t] = f1[t] \times f2[t]$ |
| v4 | 4 | $\theta^1 = [-\frac{\pi}{3}, \frac{\pi}{3})$, $\theta^2 = [-\frac{\pi}{3}, \frac{\pi}{3})$ | $f1[t] = sin(\theta_t^1),\ f2[t] = cos(\theta_t^2),\ f3[t] = 1 + (f1[t])^2$, $f4[t+1] = 1 - \frac{f1[t]}{f2[t]}$ |
| v5 | 6 | $\theta^1 = [-\frac{\pi}{3}, \frac{\pi}{3})$, $\theta^2 = [-\frac{\pi}{3}, \frac{\pi}{3})$ | $f1[t] = sin(\theta_t^1),$ $f2[t] = 1 + (f1[t])^2 + \mathcal{N}(0, \mathbf{I}) \times 10^{-4},$ $f3[t] = cos(\theta_t^3),$ $f4[t] = 1 - (f3[t])^2 + \mathcal{N}(0, \mathbf{I}) \times 10^{-4},$ $f5[t] = tan(\theta_t^5),\ f6[t] = 1 - (f5[t])^2 + \mathcal{N}(0, \mathbf{I}) \times 10^{-4}$ |
| v6 | 3 | $\theta^1 = [-\frac{\pi}{3}, \frac{\pi}{3})$, $\theta^2 = [-\frac{\pi}{3}, \frac{\pi}{4})$, $\theta^3 = [-\frac{\pi}{3}, \frac{\pi}{3})$ | $f1[t] = sin(\theta_t^1),\ f2[t] = cos(\theta_t^2),$ $f3[t+1] = 1 + f3[t] - \frac{f1[t]}{f2[t]};\ [2 \leq t \leq 100],$ $f3[1] = 1 - \frac{sin(\theta_0^1)}{cos(\theta_0^2)}$ |

Table 6: Properties of synthetic data

## A.2  Model architecture

Figure - 6 shows the detailed architecture of SADI's denoising model.

## A.3  Hyperparameter search

**MICE**, **BRITS**, and **SAITS** have only few hyper-parameters. For MICE, we found the number of iterations 30 is the optimal choice for all of the datasets. BRITS has the number of hidden state embedding size as

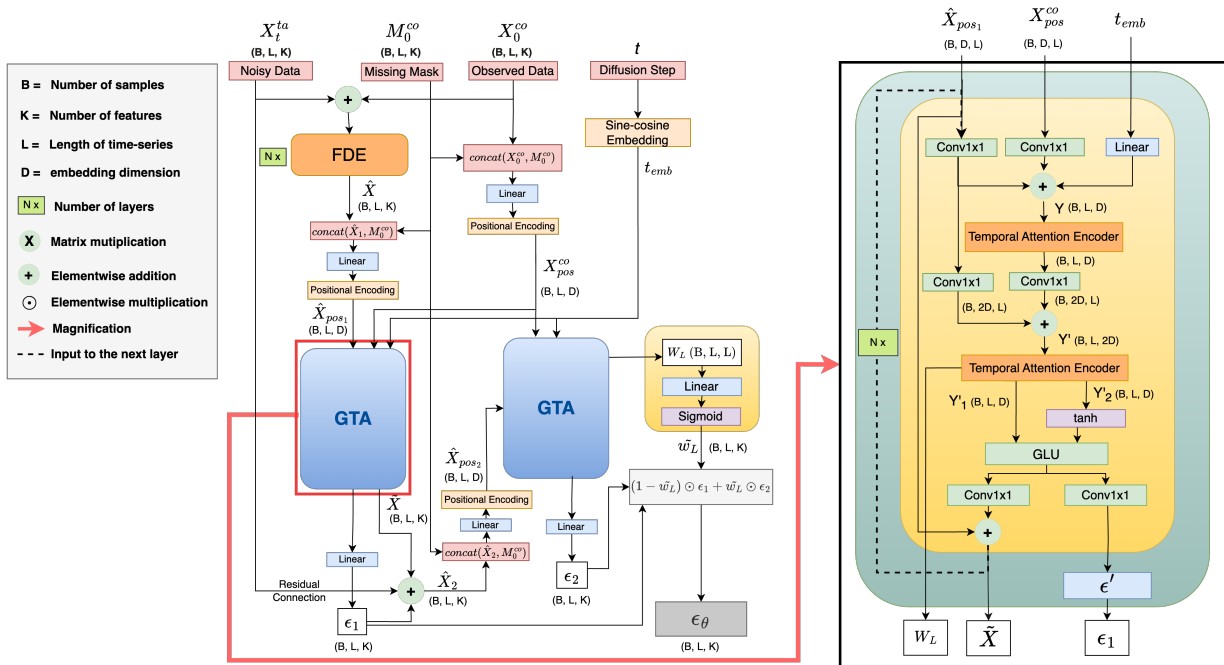

Figure 6: Detailed architecture of the denoising function of SADI

hyper-parameter and setting it to 64 works best for all the datasets. Increasing it for the larger feature size datasets did not give us better performance. For SAITS, we varied three hyper-parameters number of layers $n_{layers}$ (3, 4, 5, 6), dimension of linear embedding $d_{emb}$ (256, 512, 1024) and size of the feed forward network inside the self-attention encoder $d_{ffn}$ (128, 256, 512) for different datasets.

For **CSDI**, we employ the official implementation of CSDI (Tashiro et al. (2021)) as our baseline and search for appropriate hyperparameter settings for each dataset. Table - 7 outlines the range of hyperparameter values explored for CSDI. For most of the cases, the default hyperparameter settings provided by the authors suffice. However, for the NACSE (PRISM (2014)) and Electricity (Dua & Graff (2017)) datasets, we faced GPU RAM limitations due to their high dimensionality. Consequently, we had to limit the use of residual channels to just 4.

| Hyperparameters | Values |
|---|---|
| learning rate | 0.01, 0.001, 0.0001 |
| $\beta_0$ | 0.0001 |
| $\beta_{end}$ | 0.3, 0.5, 0.7 |
| Residual layers | 3, 4, 5 |
| Residual channels | 4, 16, 32, 64 |
| Diffusion embedding dim | 128 |
| Diffusion steps | 50, 60, 70, 100 |
| Feature embedding dim | 128 |
| Time embedding dim | 16 |
| Self-attention heads | 4, 8 |

Table 7: CSDI hyperparameters

For **SADI**, the majority of diffusion hyperparameters align with those used in CSDI. The search range for hyperparameters in SADI is detailed in Table - 8. Importantly, for the high-dimensional datasets mentioned earlier, we did not encounter the need to impose any limitations on these hyperparameters in the case of SADI.

| Hyperparameters | Values |
|---|---|
| learning rate | 0.01, 0.001, 0.0001 |
| $\beta_0$ | 0.0001 |
| $\beta_{end}$ | 0.3, 0.5, 0.7 |
| Diffusion embedding dim | 128 |
| Diffusion steps | 50, 60, 70, |
| $N_{FDE}$ | 3, 4 |
| $N_{GTA}$ | 3, 4 |
| $d_{model}$ | 128, 256, 512, 1024 |
| $d_{inner}$ | 128, 256, 512 |
| $d_k$ | 64 |
| $d_v$ | 64 |
| Self-attention heads | 8 |

Table 8: SADI hyperparameters

To ensure optimal performance, we employ a grid search technique to choose the hyperparameters for each model and dataset. The search is conducted over the parameters listed in Tables - 7 and 8. To evaluate the performance, we create artificial missingness through a random missing mask applied to the validation sets of the corresponding dataset. This missingness is then used as the ground truth for the evaluation.

## A.4 Correlation analysis with VIF

The Variance Inflation Factor (VIF) is a statistical measure used to evaluate the presence of multicollinearity within regression analysis. Multicollinearity arises when there is a strong correlation between two or more independent variables within a regression model. It is computed by the following formula. Here, $R_i^2$ denotes the unaltered coefficient of determination when regressing the $i$-th independent variable against the others. $R^2$ value for a feature is always between 0 and 1, where 0 means no-correlation and 1 means perfect correlation (a linear regression perfectly fits the observed data for the current dependent variable).

$$VIF_i = \frac{1}{1 - R_i^2}$$

We computed the VIF for our synthetic datasets (which did not contain any missing values) and two real-world datasets, specifically the Electricity and Air Quality datasets, after excluding rows with missing values. Unfortunately, we were unable to perform the same analysis for the AgAID and NACSE datasets because all rows in these datasets had at least one missing value. Table 9 presents the average VIF values for the features in each of these datasets. It is evident that within the synthetic datasets, the dataset $v1$ exhibits the lowest level of feature correlation, as the average VIF for its features is less than 5. In statistics, a VIF exceeding 5 signifies a high degree of correlation among features. The dataset **v3** shows $\infty$ VIF because features $f1$, $f2$, $f3$ have linear dependency and each of them can be perfectly predicted from the other two with a linear regression model.

## A.5 Results for random, complete blackout, and forecasting scenario

We conducted experiments aligned with existing research to assess SADI's performance relative to CSDI. Our evaluation consists of three scenarios: random missingness (at 10%, 30%, and 50%), complete blackout missing data (where all features were missing for e.g. 10, 30, and 50 consecutive time steps), and a forecasting scenario (predicting outcomes for e.g. 10, 30, and 50 future time steps). These evaluations were done for AgAID, NACSE, and Electricity datasets. In the case of the Air Quality dataset, we used the ground truth test data made available in Tashiro et al. (2021).

Table - 11, 10, and 12 present the Mean Squared Error (MSE) results along with 95% confidence interval, derived from 20 inference trials on the test sets. For the forecasting scenario, the inference setup remains

| Dataset | Avg VIF of the features |
|---|---|
| v1 | 2.12237 |
| v2 | 5.56661 |
| v3 | $\infty$ |
| v4 | 28.76032 |
| v5 | 57.37916 |
| v6 | 9.92046 |
| Electricity | 11.60544 |
| Air Quality | 47.18543 |

Table 9: Average Variance Inflation Factor of the features of different datasets

the same, so it results in a single experiment for forecasting. Here, we again observe that across all datasets SADI performs better than other existing models.

| | Random missing (%) | | |
|---|---|---|---|
| | 10% | 30% | 50% |
| MICE | 0.00656 ± 9.3e-05 | 0.01149 ± 0.0001 | 0.01673 ± 0.0001 |
| BRITS | 0.00781 ± 0.0001 | 0.01087 ± 8.8e-05 | 0.01354 ± 0.0001 |
| SAITS | 0.00734 ± 0.0001 | 0.00852 ± 5e-05 | 0.01101 ± 4e-05 |
| CSDI | 0.01913 ± 9e-05 | 0.01783 ± 6e-05 | 0.01577 ± 5e-05 |
| SADI | **0.00564 ± 7e-05** | **0.00581 ± 4e-05** | **0.00595 ± 4e-05** |
| | Blackout missing (length) | | |
| | 50 | 100 | 150 |
| MICE | 0.04968 ± 0.0152 | 0.06412 ± 0.0118 | 0.07189 ± 0.0024 |
| BRITS | 0.05865 ± 0.0093 | 0.10718 ± 0.0210 | 0.12596 ± 0.0201 |
| SAITS | 0.05905 ± 0.0143 | 0.07254 ± 0.0092 | 0.07489 ± 0.0070 |
| CSDI | 0.08860 ± 0.0231 | 0.13408 ± 0.0168 | 0.16927 ± 0.0307 |
| SADI | **0.02832 ± 0.00516** | **0.03184 ± 0.0043** | **0.03560 ± 0.0027** |
| | Forecasting (length) | | |
| | 50 | 100 | 150 |
| MICE | 0.06424 | 0.06413 | 0.06414 |
| BRITS | 0.24533 | 0.26943 | 0.26943 |
| SAITS | 0.06552 | 0.06533 | 0.06533 |
| CSDI | 0.08143 | 0.08456 | 0.08330 |
| SADI | **0.025044** | **0.02508** | **0.02468** |

Table 10: Comparison of MSE with 95% confidence interval between SADI and other 4 models in the **random** missing, **blackout** missing, and **forecasting** scenarios for **NACSE temperature** data

| | Random missing (%) | | |
|---|---|---|---|
| | 10% | 30% | 50% |
| MICE | 0.02504 ± 0.0021 | 0.03222 ± 0.0017 | 0.04679 ± 0.0019 |
| BRITS | 0.01122 ± 0.0006 | 0.01301 ± 0.0006 | 0.01726 ± 0.0006 |
| SAITS | 0.00279 ± 0.0002 | 0.00336 ±0.0002 | 0.00518 ± 0.0002 |
| CSDI | 0.00419 ± 0.0002 | 0.00462 ± 0.0002 | 0.00602 ± 0.0003 |
| SADI | **0.00026 ± 6e-05** | **0.00025 ± 4e-05** | **0.00049 ± 7e-05** |
| | **Blackout missing (length)** | | |
| | 50 | 100 | 150 |
| MICE | 0.06298 ± 0.0121 | 0.06157 ± 0.0076 | 0.0588 ± 0.0085 |
| BRITS | 0.06154 ± 0.0070 | 0.05551 ± 0.0041 | 0.05841 ± 0.0048 |
| SAITS | 0.05559 ± 0.0053 | 0.05885 ± 0.0042 | 0.06092 ± 0.0051 |
| CSDI | 0.03299 ± 0.0060 | 0.03255 ± 0.0036 | 0.03373 ± 0.0043 |
| SADI | **0.02230 ± 0.0033** | **0.02401 ± 0.0030** | **0.02752 ± 0.0044** |
| | **Forecasting (length)** | | |
| | 50 | 100 | 150 |
| MICE | 0.07654 | 0.07676 | 0.07710 |
| BRITS | 0.07757 | 0.07773 | 0.07773 |
| SAITS | 0.07745 | 0.07776 | 0.07775 |
| CSDI | 0.05694 | 0.05715 | 0.05655 |
| SADI | **0.05094** | **0.05106** | **0.05051** |

Table 11: Comparison of MSE with 95% confidence interval between SADI and other 4 models in the **random** missing, **blackout** missing, and **forecasting** scenarios for **AgAID Merlot Cultivar** data

| | Random missing (%) | | |
|---|---|---|---|
| | 10% | 30% | 50% |
| MICE | 9.0664 ± 0.0106 | 10.7183 ± 0.0233 | 12.5995 ± 0.0356 |
| BRITS | 12.0307 ± 0.0281 | 13.1066 ± 0.0064 | 14.2556 ± 0.0270 |
| SAITS | 10.9526 ± 0.0167 | 12.2415 ± 0.0090 | 13.9783 ± 0.0373 |
| CSDI | 33.4264 ± 0.3796 | 29.2097 ± 0.1018 | 20.1168 ± 0.0944 |
| SADI | **4.65954 ± 0.0076** | **4.50983 ± 0.0068** | **4.82148 ± 0.0044** |
| | **Blackout missing (length)** | | |
| | 10 | 30 | 50 |
| MICE | 31.6813 ± 0.0526 | 31.6907 ± 0.0566 | 31.6736 ± 0.0115 |
| BRITS | 21.6905 ± 0.2231 | 24.8239 ± 0.1039 | 26.0286 ± 0.1050 |
| SAITS | 32.75354 ± 0.0325 | 33.1214 ± 0.1087 | 32.6807 ± 0.0458 |
| CSDI | 22.9892 ± 0.5940 | **10.1132 ± 0.1805** | **6.50201 ± 0.0222** |
| SADI | **9.35785 ± 0.4959** | 11.20489 ± 0.1394 | 12.51583 ± 0.0918 |
| | **Forecasting (length)** | | |
| | 10 | 30 | 50 |
| MICE | 31.6291 | 31.5622 | 31.5622 |
| BRITS | 27.7793 | 18.9535 | 18.9535 |
| SAITS | 24.7884 | 23.9151 | 23.9152 |
| CSDI | 28.0036 | 68.6411 | 68.5007 |
| SADI | **4.08528** | **4.87973** | **4.89543** |

Table 12: Comparison of MSE with 95% confidence interval between SADI and other 4 models in the **random** missing, **blackout** missing, and **forecasting** scenarios for **Electricity Load Diagram** data

| | Random missing (%) | | |
|---|---|---|---|
| | 10% | 30% | 50% |
| MICE | 0.0314 ± 0.0006 | 0.0292 ± 0.0003 | 0.0285 ± 0.0002 |
| BRITS | 0.0232 ± 0.0031 | 0.0173 ± 0.0020 | 0.0191 ± 0.0014 |
| SAITS | 0.0231 ± 0.0028 | 0.0168 ± 0.0018 | 0.0184 ± 0.0012 |
| CSDI | 0.0013 ± 9.0e-05 | 0.0011 ± 5.2e-05 | 0.0014 ± 8.9e-05 |
| SADI | **0.0006 ± 0.0001** | **0.0006 ± 8.6e-06** | **0.0007 ± 3.2e-05** |
| | **Blackout missing (length)** | | |
| | 5 | 10 | 15 |
| MICE | 0.0539 ± 0.0623 | 0.0468 ± 0.0458 | 0.0417 ± 0.0372 |
| BRITS | 0.0334 ± 0.0060 | 0.0247 ± 0.0165 | 0.0326 ± 0.0227 |
| SAITS | 0.0319 ± 0.0009 | 0.0238 ± 0.0153 | 0.0309 ± 0.0219 |
| CSDI | 0.0036 ± 0.0051 | 0.0082 ± 0.0074 | 0.0080 ± 0.0046 |
| SADI | **0.0024 ± 0.0009** | **0.0051 ± 0.0044** | **0.0080 ± 0.0038** |
| | **Forecasting (length)** | | |
| | 5 | 10 | 15 |
| MICE | 0.0289 | 0.0289 | 0.0289 |
| BRITS | 0.0271 | 0.0365 | 0.0410 |
| SAITS | 0.0221 | 0.0311 | 0.0370 |
| CSDI | 0.0191 | 0.0261 | 0.0285 |
| SADI | **0.0180** | **0.0248** | **0.0265** |

Table 13: Comparison of MSE between SADI and other 4 models in the **random** missing, **blackout** missing, and **forecasting** scenarios for **Air Quality** data

