# OpenReview forum: "Self-attention-based Diffusion Model for Time-series Imputation in Partial Blackout Scenarios"
_TMLR — Rejected by TMLR_

### Review · Reviewer_sSx1 · 2024-03-06

**Summary Of Contributions:**

This work proposes SADI, “Self-Attention-based Diffusion Model for Time Series Imputation”, a new approach of diffusion-model-based imputation, under the partial blackout missing patterns wherein a subset of features remains missing for one or several consecutive time steps.

The core contribution of the proposed method is the incorporation and combination of the “Feature Dependency Encoder” and “Gated Temporal Attention” blocks, together with a two-stage imputation process. The proposed method are validated on synthetic and real datasets.

**Audience:**

Yes

**Claims And Evidence:**

Yes

**Requested Changes:**

More insights/comments on the architecture design, and how they are related to the particle blackout scenario will be extremely helpful.

“In our experiments, we observed that focusing solely on optimizing the final $\epsilon_\theta$ prediction does not lead to the training loss converging to a favorable minima.": Is there any insight into why the loss in eq (16) works better? Additionally, since $\epsilon_\theta$ is a linear combination of $\epsilon_1$ and $\epsilon_2$, does this fact have any interesting impact on the minimization of the loss function (16)?

For synthetic datasets, from the caption of Figure 3, it seems like the time series is Markov (“the t-th time step’s feature depends on the same or other features’ t−1-th time step’s value"). It would be interesting to try some non-Markov time series (with short time dependences) as well. Moreover, it seems unclear how much training data is used to train the diffusion model for the synthetic data.

**Strengths And Weaknesses:**

Strengths:

(i) The partial blackout missing mechanism is more general than commonly studied missing patterns such as 'missing at random'.
(ii) The proposed method performs better in numerical experiments.

Weakness:

There lacks enough interpretations and explanations for the proposed architecture; more insights into the numerical results and algorithm design are also needed.

---

### Review · Reviewer_ujve · 2024-03-10

**Summary Of Contributions:**

This paper presents a data imputation method based on a diffusion model. The training algorithm incorporates a random masking approach, which the authors argue makes the proposed framework more versatile than the existing missing at random strategy.

**Audience:**

Yes

**Broader Impact Concerns:**

No ethical concerns.

**Claims And Evidence:**

No

**Requested Changes:**

There have been a few recent studies employing diffusion models for data imputation. However, the authors do not appear to offer a clear discussion on the novelty.

Most of the existing Bayesian imputation methods, such as those based on Bayesian latent models, are not discussed. The authors' use of the term "generative model" seems to specifically refer to those based on deep learning. However, traditionally, it is a generic term that represents a data generating process.

The criticism of models having a narrow assumption on the missing pattern is valid. However, it is not evident what exactly the proposed solution is. It appears that the random masking mentioned in Section 5 is the solution, but I wonder if it aligns with the criticism they raised in the introduction.

**Strengths And Weaknesses:**

* The use of diffusion models, which are relatively new generative models.
* The recognition that the assumption about underlying missing patterns can be significant.

---

### Review · Reviewer_oimQ · 2024-03-29

**Summary Of Contributions:**

The paper proposes a neural network based imputation method for time-series data. The proposed method, called SADI, imputes missing values using the well-known diffusion approach. building blocks of SADI consist of three components, i.e., feature dependency encoder (FDE), gated temporal attention (GTA), and two-stage imputation process. The authors claim that SADI can incorporate feature and temporal dependency jointly unlike existing methods, and that can deal with a variety of missing patterns, called 'partial blackout' for which examples are shown in Fig1. The empirical performance is verified through six synthetic datasets and four real datasets.

**Audience:**

Yes

**Claims And Evidence:**

Yes

**Requested Changes:**

Technical descriptions are sometimes unclear. Could you provide more detailed elaboration?

- Fig2 (diagram of the model) is informative. On the other hand, I still do not fully understand the meaning of the claim 'SADI models feature and time dependencies jointly, thereby capturing the combined correlations of both features and time', because in Fig2, both of SADI and the existing method sequentially apply components for capturing feature dependency and temporal dependency. Why can only SADI be interpreted as 'jointly' modeling two types of dependency. The partitioning in CSDI causes some critical issue about the joint modeling?

- Related to the above question, the discussion on partial blackout is not fully convincing. How this concept is essentially connected to the formulation of the proposed method? Many existing methods can deal with general pattern of missing positions (and as described above, both the proposed and existing method, sequentially (not simultaneously) apply models for feature and time dependency, respectively). What is advantage of SADI compared with existing methods in terms of diversity of missing value patterns? The authors emphasize partial blackout mainly in introduction, but almost nothing is discussed in the methodology section (Sec 4).

- The proposed method sequentially applies feature dependency encoder and temporal attention. Then, temporal correlation that depends on feature can be incorporated. On the other hand, feature correlation that depends on time can be incorporated in this sequential modeling?

- The advantage compared with SAITS is not sufficiently described, though the architecture is quite similar to the proposed method.

- How the artificial missing data is created? Created based on patterns in Fig1? Dose the selection of patterns for the training set have an effect on the final performance? Intuitively, the model is seemingly good at patterns that is included in the training dataset a lot, while patterns that are not sufficiently included in the training set may not be predicted accurately. Even when a model architecture has an ability of handling temporal- and feature- dependency jointly, it may not be optimized if the target in the training dataset can be predicted without using such joint dependency. In this sense, way of defining artificially missing data may have impact on performance of partial blackout setting.

- Why 'No 2nd block' shows a significant decrease in the ablation study Table 5? On the other hand, CSDI has stable performance even it is with one feature and time dependency block. Further, compared with other existing methods in Table 1, performance in table 5 (three methods defined by removing one component from SADI as an ablation) is often much worse. The authors seemingly argue that the superiority of the proposed method mainly stems from simultaneously modeling of feature and time dependency. For me, it is not clear whether this contradicts the fact that simply removing one component from SADI dramatically worsens performance.

- How was the hyper-parameters were selected? In A.3, candidates of parameters are written, but I cannot find the criterion and which data (artificially missing data?) is used for the selection.

- In 'Other methods' of Section 2, each method is just enumerated. Pros and cons are not clear for readers.

Showing some example plots of synthetic and/or real datasets would be informative for readers.

Minor issues:

- The numerical format of tables (such as Table 1) is sometimes inconsistent. For example, 'v4 2 4' of CSDI in Table 1 is written as 2.3e-04 while 'v4 2 3' is written as 0.0002. Not only here, similar inconsistencies exist.

- In line 8 of Sec5.2, sin(\theta^i_t t) is sin(\theta^i_t) ?

**Strengths And Weaknesses:**

S: The topic (imputation for time-series data) should be important, and the experimental results indicate superior prediction performance.

W: Descriptions of novelty on partial blackout is unclear. In introduction and abstract, variations of missing value patterns (partial blackout) are explained in detail, but an essential connection between these missing patterns and the proposed method is not fully clarified in the main text.

W: As described below (in requested changes), I am currently not fully convinced by the basic rationale behind why the proposed method can capture feature and temporal dependency jointly. Although this should be a main technical claim, I do not think the current justification is sufficient, particularly about difference from existing methods.

---

### Decision · Action_Editor_w16r · 2024-05-17

**Recommendation:** Reject

**Comment:**

This paper introduces SADI, a neural network-based method for time-series imputation using a diffusion model. SADI integrates feature and temporal dependencies with a two-stage imputation process, effectively handling partial blackout missing patterns. The method demonstrates strong performance on both synthetic and real datasets.

In this paper, the paper claims three contributions.
1. Exploration of partial blackout scenario
2. Explicit modeling of feature and temporal correlations and two-stage imputation
3. Show the SOTA performance in empirical study

The main concern is that the main technical claims are not fully justified. 1) Drastic performance deterioration in the ablation setting. More specifically, in the ablation study, the performance is significantly degraded if the 2nd block is removed, where the 2nd block is to refine the model. This indicates that the second block is important and the feature and temporal correlations section is not that super useful. The explicit modeling of feature and temporal correlation is one of the key claims.  The claim should be carefully verified by more experimental studies. For example, changing the feature-temporal correlation modeling part with an existing component and then applying the imputations. If this combination significantly degrades the performance in the end, it means that the first component is important with the combination of the proposed imputation.  Moreover, the paper lacks training on partial blackout. Since partial blackout scenario is in the main claim, more empirical study with careful discussion is needed.

Thus, at this point, it is difficult to accept the paper. I encourage authors to revise the paper and resubmit it to another venue.

**Audience:**

Diffusion models are an important topic in ML. It may attract some researchers in the community.

**Claims And Evidence:**

Claims are mostly addressed by the response. However, the main concern of a reviewer is not fully addressed. 1) Drastic performance deterioration in the ablation setting. 2)Absence of training on partial blackout.